# Relaxing Accurate Initialization Constraint for 3D Gaussian Splatting

## Abstract

In this work, we investigate the limitations of the 3D Gaussian Splatting (3DGS) optimization scheme, revealing why it undergoes significant performance drops when initialized with noisy or random point clouds. Through in-depth analysis, we identify a key limitation of the 3DGS optimization: limited Gaussian transportability. Since Gaussians are optimized solely based on image photometric loss, the optimization tends to overfit the parameters of the projected Gaussians to improve reconstruction at their current positions, rather than relocating them to more optimal locations. This leads to producing under-reconstructed regions when starting with noisy or random initialization, failing to transport Gaussians to correct locations. Based on our findings, we propose **RAIN-GS** (**R**elaxing **A**ccurate **IN**itialization Constraint for 3D **G**aussian **S**platting), a set of simple yet effective modifications, including initializing sparse Gaussians with large variances, progressive Gaussian low-pass filtering, and an Adaptive Bound-Expanding split algorithm. These modifications enable Gaussians to effectively redistribute across the scene, capturing both coarse structure and fine details. By addressing the inherent limitations of 3DGS, RAIN-GS allows effective training even with random point clouds, significantly enhancing reconstruction quality.

## 1 Introduction

Novel view synthesis is one of the essential tasks in computer vision and computer graphics, aiming to render novel views of a 3D scene given a set of images. It has a wide range of applications in various fields, including augmented reality and virtual reality (Xu et al., 2023), robotics (Adamkiewicz et al., 2022), and data generation (Ge et al., 2022). Recently, neural radiance fields (NeRFs) (Mildenhall et al., 2021) and 3D Gaussian splatting (3DGS) (Kerbl et al., 2023) have demonstrated remarkable success in this task, where 3DGS further pushes the boundary of real-time rendering through explicitly representing the scene with Gaussians.

Despite its remarkable results, compared to NeRFs, 3DGS requires an additional input of initial point cloud. In addition, the quality of the initial point cloud is one of the essential requirements of 3DGS, showing large performance drops when trained with randomly initialized point cloud (Kerbl et al., 2023). To mitigate such performance degradation, 3DGS and its extensions (Yu et al., 2023; Luiten et al., 2024) often utilize Structure-from-Motion (SfM) (Schonberger & Frahm, 2016) algorithms, which provide both accurate camera poses and point clouds.

However, in real-world scenarios, SfM can also fail to produce accurate point clouds, such as in scenes with symmetry, textureless regions, and dynamic movements inducing occulsions (Bian et al., 2023; Zhang et al., 2022). In addition, instead of applying SfM algorithms, camera poses are often estimated with external sensors (Geiger et al., 2013; Sturm et al., 2012) or pre-defined as given trajectories as in text- or image-to-3D generation (Tang et al., 2023; Yi et al., 2024). Initial point clouds become unavailable in these scenarios, which leads to performance degradation in 3DGS.

To understand this strict requirement of accurate point clouds, which has not yet been fully explored, in this work, we start with a natural question: *"Why is accurate initial point cloud so important for 3D Gaussian Splatting?"*. By conducting an in-depth analysis, we reveal an important limitation of the current 3DGS optimization scheme: limited Gaussian transportability. This is primarily due to the Gaussian being optimized solely with image photometric loss, which fails to provide clear guidance for the Gaussians to move to their optimal positions. As a result, the optimization process

often leads to under-reconstruction of the scene. We further reveal that this problem has simply been less highlighted with SfM point clouds as they already provide information about where the scene geometry exists, reducing the need for Gaussian transportation.

Based on our analysis, we propose a simple yet effective method, **RAIN-GS** (**R**elaxing **A**ccurate **IN**itialization Constraint for 3D **G**aussian **S**platting), composed of simple modifications to address the existing limitation of 3DGS. Specifically, we initialize sparse Gaussians with large variance, employ progressive Gaussian low-pass filtering, and split Gaussians with a new Adaptive Bound-Expanding split algorithm, which enables the Gaussians to effectively redistribute across the scene, capturing both coarse structure and fine detail throughout the optimization. RAIN-GS effectively mitigates the limitation of the original 3DGS optimization, enabling 3DGS to achieve high-quality reconstructions even with random initializations.

In summary, our main contributions are as follows:

- We conduct an in-depth analysis and identify the key limitation of the current 3DGS optimization scheme: limited Gaussian transportability. This is due to the 3DGS optimization's sole reliance on image photometric loss, which fails to provide clear guidance for the Gaussians to move to their optimal positions.

- While limited Gaussian transportability is an inherent limitation of 3DGS optimization, we further reveal that as accurate initializations provide information of where the scene geometry exists, this problem has been less highlighted.

- Based on our findings, we propose **RAIN-GS**, which effectively enables the Gaussians to redistribute across the scene, achieving on-par or better reconstruction results even with random initializations.

## 2 RELATED WORK

**Structure-from-Motion (SfM).** SfM techniques (Agarwal et al., 2011; Schonberger & Frahm, 2016) have been one of the most widely used algorithms to reconstruct a 3D scene. Through iterative feature matching and bundle adjustment, SfM algorithms estimate the camera pose and point cloud of the reconstructed scene. Despite the effectiveness of SfM algorithms, its incremental nature and the computational intensity of bundle adjustment significantly increase its time complexity, often to $O(n^4)$ with respect to $n$ cameras involved (Wu, 2013). To mitigate such limitations, recent methods propose to replace the components of SfM algorithms with learnable modules (Wang et al., 2024b;a; Pan et al., 2024), accelerating the overall process.

**Neural radiance fields (NeRF).** NeRF (Mildenhall et al., 2021) has succeeded in significantly boosting the performance of novel view synthesis by optimizing an MLP that can estimate the density and radiance of any continuous 3D coordinate. With the camera poses of the given images, NeRF learns the MLP by querying dense points along randomly selected rays, which outputs the density and color of each of the queried coordinates. Various follow-ups (Barron et al., 2021; 2022; Du et al., 2023; Hong et al., 2023; Li et al., 2023; Müller et al., 2022; Song et al., 2024; Yang et al., 2023) adopted NeRF as their baseline model and further extend the ability of NeRF to model unbounded or dynamic scenes (Barron et al., 2021; 2022; Li et al., 2023), lower the required number of images for successful training (Song et al., 2024; Yang et al., 2023), or utilize an external hashgrid to accelerate the overall optimization process (Müller et al., 2022). Although all of these works show compelling results, the volume rendering from dense points along multiple rays makes NeRF hard to apply in real-time settings achieving lower rendering rates of under < 1 fps.

**3D Gaussian splatting (3DGS).** Departing from implicit representations of NeRF, 3DGS (Kerbl et al., 2023) represents the scene with explicit 3D Gaussians, achieving real-time rendering speed of over > 90 fps. Thanks to its efficiency, 3DGS has gained massive attention and has been extended to modeling large-scale scenes (Kerbl et al., 2024), dynamic scenes (Luiten et al., 2024), and enabling the training with multi-scale images (Yu et al., 2023). Nevertheless, 3DGS is not without limitations as the performance largely deteriorates when trained with sub-optimal (noisy, sparse, random) point clouds.

## 3 PRELIMINARY: 3D GAUSSIAN SPLATTING

In this section, we briefly explain 3DGS (Kerbl et al., 2023), which represents the scene with multiple 3D Gaussians (Zwicker et al., 2002). Each $i$-th Gaussian $G_i$ represents the scene with the following attributes: a position vector $\mu_i \in \mathbb{R}^3$, an anisotropic covariance matrix $\Sigma_i \in \mathbb{R}^{3\times3}$, spherical harmonic (SH) coefficients (Yu et al., 2021; Müller et al., 2022), and an opacity logit value $\alpha_i \in [0, 1)$. With these attributes, each Gaussian $G_i$ is defined in the world space $x$ as follows:

$$G_i(x) = e^{-\frac{1}{2}(x-\mu_i)^T \Sigma_i^{-1}(x-\mu_i)}. \tag{1}$$

To render an image from a pose represented by the viewing transformation $W$, the projected covariance $\Sigma'_i$ is defined as follows:

$$\Sigma'_i = JW\Sigma_i W^T J^T, \tag{2}$$

where $J$ is the Jacobian of the local affine approximation of the projective transformation. The 2D covariance matrix is simply obtained by skipping the third row and column of $\Sigma'_i$ (Zwicker et al., 2002). Finally, to render the color $C(p)$ of the pixel $p$, 3DGS utilizes alpha blending according to the Gaussians depth. For example, when $N$ Gaussians are sorted by depth, the color $C(p)$ is calculated as follows:

$$C(p) = \sum_{i=1}^{N} c_i \alpha_i G'_i(p) \prod_{j=1}^{i-1}(1 - \alpha_j G'_j(p)), \tag{3}$$

where $c_i$ is the view-dependent color value of each Gaussian calculated with the SH coefficients, and $G'_i$ is the 3D Gaussian projected to the 2D screen space.

During optimization, 3DGS adaptively adjusts the number of Gaussians through cloning and splitting, to adjust the scene from being under-/over-reconstructed. Specifically, a Gaussian is cloned in the mean position of the original Gaussian, if the scene is under-reconstructed. This can happen if the scene needs to be represented with more Gaussians and the covariance of the current Gaussian is too small. In contrast, if the scene needs to be represented with more detail and the covariance of the current Gaussian is too large, the Gaussian undergoes splitting, where the mean positions of the new Gaussians are sampled from the probability density function of the original Gaussian.

## 4 MOTIVATION

In this section, we present an in-depth analysis of the 3D Gaussian Splatting (3DGS) optimization process (Kerbl et al., 2023), focusing on the impact of different initial point cloud qualities. We begin by conducting both quantitative and qualitative comparisons using various point cloud initialization which is shown in Section 4.1. From this comparison, we reveal two characteristics of 3DGS: **1)** 3DGS heavily depends on accurate initialization, showing large performance drops even with little noise, and **2)** As the initialization becomes more inaccurate, 3DGS suffers from the scene being under-reconstructed which results in particular objects in the scene being left un-reconstructed. To further understand this behavior, we conduct a deeper analysis of the 3DGS optimization scheme in Section 4.2, which reveals a critical limitation: limited Gaussian transportability. We show that this limitation is the primary cause of the scene being under-reconstructed.

### 4.1 ANALYSIS OF VARIOUS POINT CLOUD INITIALIZATIONS

To analyze the relationship between the accuracy of initial point clouds and performance, we perform quantitative and qualitative comparisons using various initializations in 3DGS. In addition to point clouds from SfM and random initialization, we also introduce the noisy SfM initialization setting where we perturb the positions of the SfM initialized point cloud by adding a small noise. Noisy SfM is introduced to mimic the situations where the point cloud from SfM algorithms is not perfect (e.g., textureless regions, dynamic movements) [1].

Specifically, we train 3DGS on the Mip-NeRF360 (Barron et al., 2022), Tanks&Temples (Knapitsch et al., 2017), and Deep Blending Hedman et al. (2018) dataset with SfM, noisy SfM, and random point clouds. For SfM point clouds, we use the estimated point clouds from COLMAP (Schonberger & Frahm, 2016). For noisy SfM point clouds, we perturb the point cloud achieved from COLMAP

---

[1] In addition, as noisy SfM with a very large noise is similar to the random case, this shows the tendency of the performance of 3DGS to the amount of noise in the initial point cloud.

| Initial point clouds | Mip-NeRF360 | | | Tanks&Temples | | | Deep Blending | | | Average | | |
|---|---|---|---|---|---|---|---|---|---|---|---|---|
| | PSNR↑ | SSIM↑ | LPIPS↓ | PSNR↑ | SSIM↑ | LPIPS↓ | PSNR↑ | SSIM↑ | LPIPS↓ | PSNR↑ | SSIM↑ | LPIPS↓ |
| SfM | 27.462 | 0.814 | 0.219 | 23.142 | 0.841 | 0.183 | 29.623 | 0.900 | 0.251 | 26.742 | 0.852 | 0.218 |
| Noisy SfM | 27.004 | 0.799 | 0.243 | 22.592 | 0.816 | 0.219 | 29.515 | 0.899 | 0.256 | 26.370 | 0.838 | 0.239 |
| Random | 25.893 | 0.764 | 0.273 | 21.862 | 0.795 | 0.227 | 29.523 | 0.897 | 0.257 | 25.759 | 0.819 | 0.252 |

Table 1: **Quantitative comparison on Mip-NeRF360 (Barron et al., 2022), Tanks&Temples (Knapitsch et al., 2017), and Deep Blending (Hedman et al., 2018) dataset using different initial point clouds.** The results show that 3DGS heavily depends on the accuracy of initial point clouds, showing performance drops when trained with noisy SfM and random point clouds.

by adding noise [2] sampled from the Normal distribution $\epsilon \sim N(0, 0.5)$. For random point clouds, we follow previous approaches (Kerbl et al., 2023; Kheradmand et al., 2024) where points are randomly sampled from the bounding box defined by three times the bound calculated with camera poses.

The quantitative comparison shown in Table 1 indicates that 3DGS heavily depends on accurate initialization for point clouds, where small noise in the initial point cloud (noisy SfM) can also lead to large performance drops. The qualitative comparison shown in Figure 1, further identifies the primary cause of performance degradation, which is mainly due to under-reconstruction. When compared to (a) and (b), the house in the background remains missing in (c) and (d) (visualized in the red bounding box).

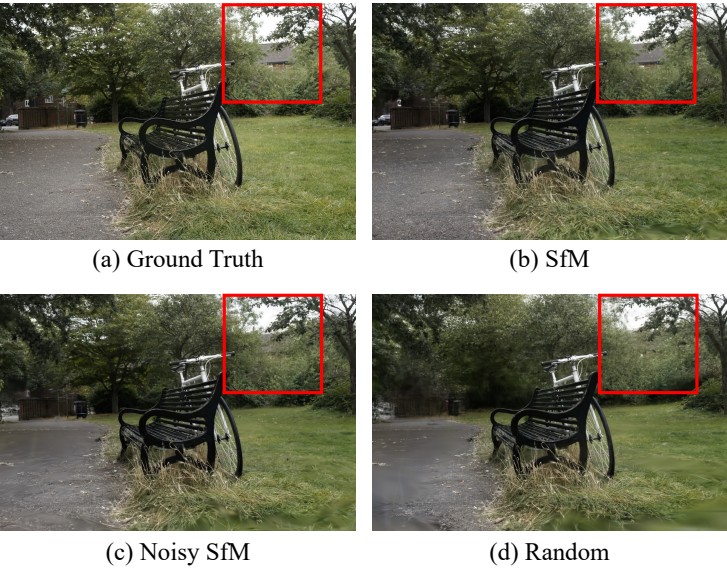

| | |
|---|---|
| (a) Ground Truth | (b) SfM |
| (c) Noisy SfM | (d) Random |

Figure 1: **Qualitative comparisons on 'bicycle' scene rendered using different initial point clouds.** The red-bounding box region shows examples of under-reconstruction, where the house in the background remains un-reconstructed.

## 4.2 ANALYSIS OF 3D GAUSSIAN SPLATTING OPTIMIZATION

As 3DGS (Kerbl et al., 2023) represents the scene using explicit 3D Gaussians, under-reconstruction can occur if Gaussians are absent or insufficient in regions where the scene geometry exists. However, when the scene is under-reconstructed due to the lack of sufficient Gaussians, 3DGS inherently has the ability to mitigate this issue by increasing the number of Gaussians through cloning. Therefore, we first hypothesize that 3DGS lacks the ability to effectively transport Gaussians, making the scene under-reconstructed due to the absence of Gaussians.

To understand this behavior, we begin with revisiting the analysis of pixelSplat (Charatan et al., 2023), where they mention the proneness of 3DGS falling into local minima due to two main reasons. **1)** The Gaussians can only receive gradients close to their means, mostly from the range not

---

[2]Note that this value is very small when compared to the initial range of SfM point clouds. A detailed explanation can be found in Section A.1 of the Appendix.

exceeding the distance of a few standard deviations, and **2)** there is no existing path for the Gaussians that will decrease the loss monotonically. As Gaussians in local minima cannot move to other locations, they become trapped and unable to explore and cover under-reconstructed regions.

However, their analysis is not directly applicable to our setting, as their analysis is limited to settings where 3DGS is trained without any per-scene optimization and without the cloning and splitting method. Therefore, we extend their analysis to the per-scene optimization setting which has not yet been explored. Specifically, we evaluate the total transportation distance of each of the Gaussians, including the movement caused by cloning and splitting. We keep the amount of movement each Gaussian takes every iteration, where we sum all the movements in length until the end of training [3].

We evaluate the total movement of the points specifically on the Mip-NeRF360 (Barron et al., 2022) dataset, where the average scene bound [4] is approximately $92 \times 53 \times 95$. As SfM point clouds already contain the information about where the scene geometry is located, starting from noisy SfM or random point clouds requires more transportation during optimization. However, as shown in Table 2, starting from SfM point clouds

|        | SfM    | Noisy SfM | Random |
|--------|--------|-----------|--------|
| Means  | 0.704  | 0.650     | 0.395  |
| Stds   | 2.207  | 0.729     | 0.402  |
| Top 1% | 10.755 | 3.646     | 1.923  |

Table 2: **Movement of Gaussians.**

results in the most transportation, revealing the lack of ability of 3DGS to relocate Gaussians to correct locations through optimization.

In addition, we also show the visualization of Gaussians before and after training. Specifically, we show the Gaussians on the 'Truck' scene of the Tanks&Temples (Knapitsch et al., 2017) datasets trained with SfM, noisy SfM, and random point clouds in Figure 2. This visualization also verifies that starting with SfM results in Gaussians moving the most. This indicates that if the Gaussians are not initialized close to where scene geometry exists, the cloning process of ADC is not sufficient to resolve all under-reconstruction scenarios. This effectively verifies our hypothesis and highlights the need for an additional method that can effectively transport the Gaussians.

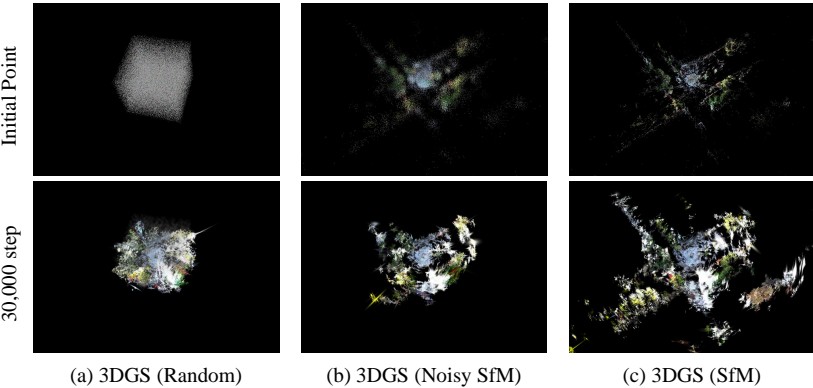

(a) 3DGS (Random)  (b) 3DGS (Noisy SfM)  (c) 3DGS (SfM)

Figure 2: **Visualization of Gaussians before and after training.** The visualization shows the position of the Gaussians before and after training done in the 'truck' scene of the Tanks&Temples dataset using different initialization (SfM, Noisy SfM, and random). The visualization indicates that SfM shows the most difference, whereas starting from random initialization shows almost no difference.

## 5 METHODOLOGY

Based on our findings revealed from the in-depth analysis of 3DGS in Section 4, we propose a simple yet effective baseline strategy **RAIN-GS** (**R**elaxing **A**ccurate **IN**itialization Constraint for 3D **G**aussian **S**platting). This strategy mainly focuses on alleviating the current limitation of the 3DGS optimization scheme, namely the limited ability to transport Gaussians which leads to under-reconstructed scenes. Specifically, **RAIN-GS** consists of three main components: **1)** Sparse-Large-

---

[3]A detailed explanation of how this experiment is conducted can be found in Section A.2 of the Appendix.

[4]The scene bound is calculated by the bound of SfM point clouds. The bound value for each scene can be found in the Table 8 in Appendix.

Variance (SLV) initialization (Section 5.1), **2)** Progressive Gaussian low-pass filter (Section 5.2), and **3)** Adaptive Bound-Expanding Split (ABE-Split) algorithm (Section 5.3).

## 5.1 SPARSE-LARGE-VARIANCE (SLV) INITIALIZATION

As discussed in Section 4.2, Gaussians easily fall into local minima due to receiving gradients from a very local region and become stuck, lacking the ability to move to other locations. To prevent the Gaussians from falling into local minima, we propose a simple yet effective modification to the original random initialization method of 3DGS (Kerbl et al., 2023). Specifically, we follow the initialization of the Gaussian parameters where covariance is determined by the distances to the three nearest neighbors but significantly reduce the initial number of Gaussians from $N = 100,000$ to $N = 10$. Despite being implementable with a simple one-line code change, this modification leads to several key improvements.

By reducing the number of initial Gaussians, the average distance between neighboring Gaussians becomes substantially larger, resulting in increased initial covariance values. This leads to what we call Sparse-Large-Variance (SLV) Initialization, wherein the Gaussians are initialized with greater spatial coverage. These larger Gaussians project to cover broader regions in the image plane, thereby receiving gradient information from larger regions during optimization. Consequently, they are more capable of learning the global structure of the scene during the optimization process, effectively mitigating scenarios where the Gaussians are overfitted to represent very local regions.

In addition to learning from larger regions, SLV initialization also provides benefits in transporting Gaussians to further locations via the splitting process. Since the splitting of Gaussians is performed by sampling from the probability density function (PDF) of the original Gaussian parameters, initializing with a larger variance naturally encourages the newly split Gaussians to explore a wider spatial area. This helps mitigate the issue of Gaussians being unable to move from local minima, allowing them to better cover the scene. Thus, SLV enhances the ability of Gaussians to adaptively explore the scene throughout the optimization process, effectively mitigating the Gaussians from falling into local minima.

## 5.2 PROGRESSIVE GAUSSIAN LOW-PASS FILTERING

Although our SLV initialization method is effective, we find that after multiple densification steps, the number of 3D Gaussians increases exponentially due to the adaptive density control, which can collapse into similar problems with the original random initialization. In order to ensure the Gaussians to receive gradients from a sufficiently large area during the optimization step, we propose a novel progressive control of the Gaussian low-pass filter which is utilized in the rendering stage.

**Gaussian low-pass filter for 3DGS.** In the rendering stage of 3DGS, the 2D Gaussian $G'_i$ projected from a 3D Gaussian $G_i$ is defined as follows:

$$G'_i(x) = e^{-\frac{1}{2}(x-\mu'_i)^T \Sigma'^{-1}_i (x-\mu'_i)}. \tag{4}$$

However, directly using projected 2D Gaussians can lead to visual artifacts when they become smaller than the size of a single pixel (Kerbl et al., 2023; Yu et al., 2023). To ensure coverage of at least one pixel, (Kerbl et al., 2023) enlarge the 2D Gaussian's scale by adding a small value to the covariance's diagonal elements as follows:

$$G'_i(x) = e^{-\frac{1}{2}(x-\mu'_i)^T (\Sigma'_i + sI)^{-1}(x-\mu'_i)}, \tag{5}$$

where $s$ is a pre-defined value of $s = 0.3$ and $I$ is an identity matrix. This process can also be interpreted as the convolution between the projected 2D Gaussian $G'_i$ and a Gaussian low-pass filter $h$ (mean $\mu = 0$ and variance $\sigma^2 = 0.3$) of $G'_i \otimes h$, which is shown to be an essential step to prevent aliasing (Zwicker et al., 2002). After applying convolution with the low-pass filter, the area of the projected Gaussian $G'_i$ is approximated by a circle. The radius of this circle is defined by three times the larger eigenvalue from the 2D covariance matrix $(\Sigma'_i + sI)$[5].

**Progressive low-pass filter control.** Instead of using a fixed value of $s$ through the entire optimization process, we notice that this value $s$ can ensure the minimum area each Gaussians have to

---

[5]We provide a detailed proof in Section B.1 in the Appendix.

cover in the screen space. As Gaussians only receive gradients inside the range of a few standard deviations (Charatan et al., 2023), learning from wider areas is essential for the Guassians to receive sufficient gradients. Therefore, to ensure the Gaussians to receive sufficient amount of gradient during training, we control $s$ to regularize the Gaussians to cover wider areas during the early stage of training and progressively learn from a more local region. Specifically, as the value $s$ ensures the projected Gaussians area to be larger than $9\pi s^6$, we define $s$ as $s = HW/9\pi N$, where $N$ indicates the number of Gaussians and $H, W$ indicates the height and width of the image respectively.

### 5.3 ADAPTIVE BOUND-EXPANDING SPLIT (ABE-SPLIT) ALGORITHM

The ABE-Split method is a straightforward extension to the original splitting algorithm in 3DGS, designed to address scenarios where under-reconstruction occurs due to the absence of Gaussians. In the original algorithm, two new Gaussians are sampled locally from the PDF of an existing Gaussian, which limits the ability to address globally under-reconstructed regions. Although SLV initialization partially addresses this problem by enabling the Gaussians to be redistributed to larger regions, we find that after multiple splitting steps, the variance of the Gaussian becomes small, where new Gaussians can be only placed locally. To overcome this limitation, we propose ABE-Split, where an additional Gaussian is split during the early stages of optimization. Specifically, we initialize a third Gaussian at a position outside the current bounds defined with the positions of the Gaussians, by multiplying a scalar to the current Gaussian coordinate. Although this approach is extremely simple, when combined with our SLV initialization, we can effectively expand the bounds of the Gaussians, ensuring the Gaussians to be actively re-distributed in globally under-reconstructed areas.

## 6 EXPERIMENTS

### 6.1 IMPLEMENTATION DETAILS

We implement our model based on the 3DGS (Kerbl et al., 2023). We follow the same training process of the existing implementation in all datasets. For our sparse-large-variance (SLV) random initialization, we set the initial number of Gaussians to $N = 10$. For progressive low-pass filter control, we find that re-defining the value $s$ as $s = \min(\max(HW/9\pi N, 0.3), 300.0)$ every 1,000 steps results in better results compared to changing the value every step and adopt this strategy as default. For training SH coefficients, we set the maximum degree as 3 following the original implementation. As we regularize the Gaussians to learn from larger regions to receive a sufficient amount of gradient, we lower the divide factor from 1.6 to 1.4. In addition, as spherical harmonics should be learned with higher degrees when the Gaussians are modeling local regions, we increase the SH degree after 5,000 steps which is approximately when the low-pass filter value becomes $s = 0.3$. All other hyperparameters are left unchanged.

### 6.2 DATASETS

We conduct experiments on multiple datasets, including experiments on the dataset where the initial point cloud is not accessible. Specifically, we use Mip-NeRF360 (Barron et al., 2022), Tanks&Temples (Knapitsch et al., 2017), and Deep Blending (Hedman et al., 2018) dataset previously utilized in 3DGS (Kerbl et al., 2023). For the evaluation of these datasets, we follow the evaluation protocol of 3DGS, where every 8th image is used as the test set and outdoor images and indoor images of the Mip-NeRF360 dataset are downscaled by the factor of four and two respectively. We further conduct experiments on the RealEstate-10K (Re10K) dataset (Zhou et al., 2018), to demonstrate the effectiveness of our strategy when initial point clouds are not accessible [7]. For Re10K, every 8th image is also used as the test set without downscaling the images.

### 6.3 BASELINES

We compare against 3DGS, Mip-Splatting (Yu et al., 2023), 2D Gaussian splatting (Huang et al., 2024). We also compare with 3DGS trained from random initialization. For 3DGS, as their public code shows slightly better performance compared to their reported values, we show both the reported values (3DGS) and the values achieved from their public code (3DGS (re-run)).

---

[6]We provide a detailed proof in Section B.2 in the Appendix.

[7]Re10K only provides the camera poses of the images estimated from ORB-SLAM (Mur-Artal et al., 2015)

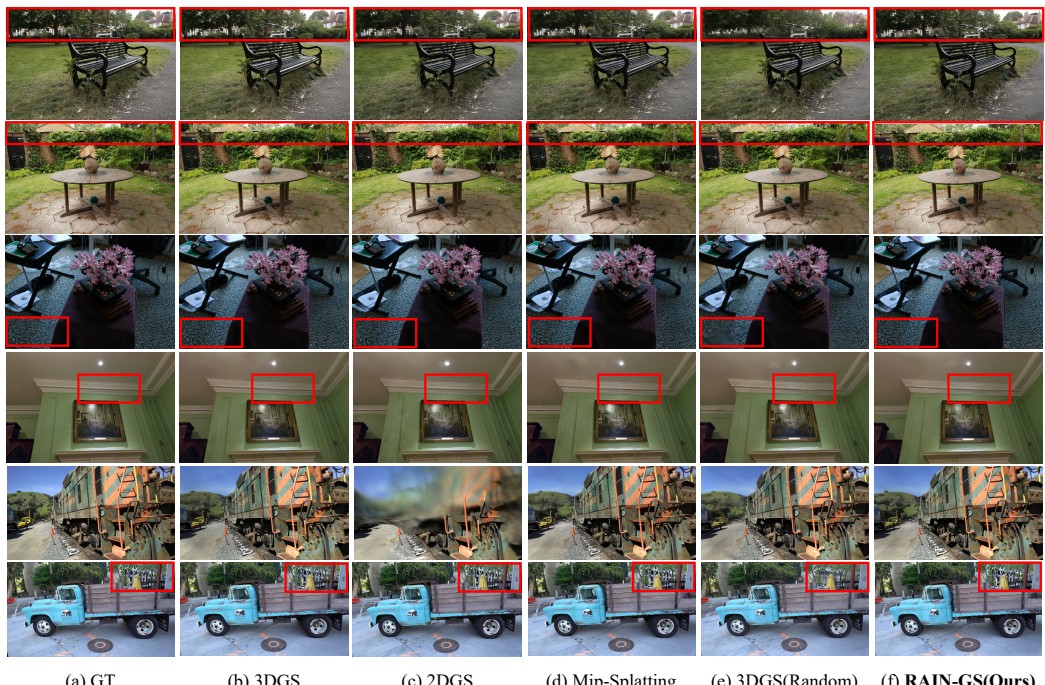

Figure 3: **Qualitative results on Mip-NeRF360 (Barron et al., 2022), Tanks&Temples (Knapitsch et al., 2017) and Deep Blending (Hedman et al., 2018) datasets.** We compare the results with both methods that utilize SfM point clouds ((b),(c),(d)) and the method that uses random point clouds ((e),(f)). Unlike (e), which shows under-reconstructed regions, our method (g) effectively captures missing details.

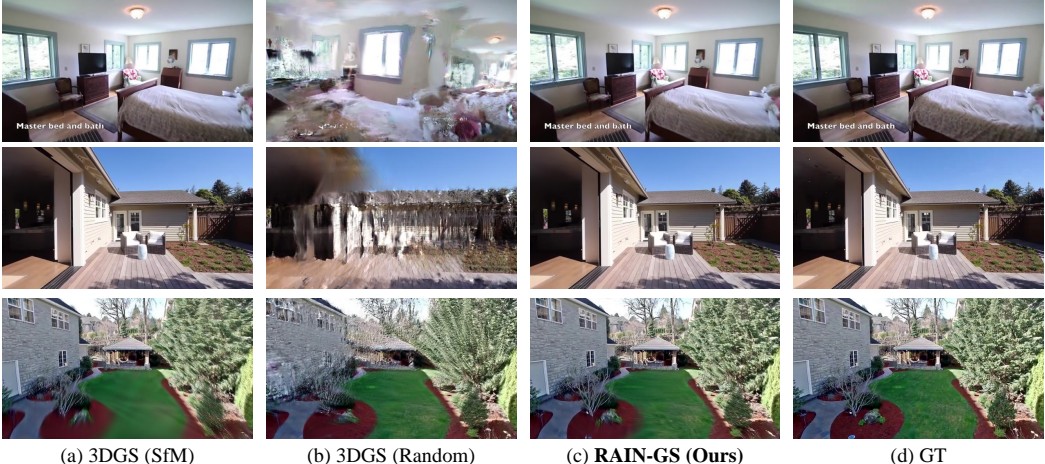

Figure 4: **Qualitative results on RealEstate-10K (Zhou et al., 2018) dataset.** We compare the results with 3DGS trained from SfM and random point clouds. As RE10K does not provide initial point clouds, we have preprocessed COLMAP (Schonberger & Frahm, 2016) to train 3DGS (SfM). Note that although Ours and 3DGS (Random) is not trained with the COLMAP poses, Ours show competitive results with 3DGS (SfM).

For RealEstate-10K, we specifically compare against 3DGS trained from SfM and random initialization. As Re10K does not provide any initial point clouds, to evaluate 3DGS from SfM, we have pre-processed the images with COLMAP (Schonberger & Frahm, 2016). Note that Ours and 3DGS from random point clouds utilize the camera poses from the dataset without further refinement.

| Method | SfM points | Mip-NeRF360 Outdoor Scene | | | | | | | | | | | | | | |
| | | bicycle | | | flowers | | | garden | | | stump | | | treehill | | |
| | | PSNR↑ | SSIM↑ | LPIPS↓ | PSNR↑ | SSIM↑ | LPIPS↓ | PSNR↑ | SSIM↑ | LPIPS↓ | PSNR↑ | SSIM↑ | LPIPS↓ | PSNR↑ | SSIM↑ | LPIPS↓ |
|---|---|---|---|---|---|---|---|---|---|---|---|---|---|---|---|---|
| Plenoxels Yu et al. (2021) | ✗ | 21.912 | 0.496 | 0.506 | 20.097 | 0.431 | 0.521 | 23.495 | 0.606 | 0.386 | 20.661 | 0.523 | 0.503 | 22.248 | 0.509 | 0.540 |
| INGP-Base Müller et al. (2022) | ✗ | 22.193 | 0.491 | 0.487 | 20.348 | 0.450 | 0.481 | 24.599 | 0.649 | 0.312 | 23.626 | 0.574 | 0.450 | 22.364 | 0.518 | 0.489 |
| INGP-Big Müller et al. (2022) | ✗ | 22.171 | 0.512 | 0.446 | 20.652 | 0.486 | 0.441 | 25.069 | 0.701 | 0.257 | 23.466 | 0.594 | 0.421 | 22.373 | 0.542 | 0.450 |
| 3DGS (Kerbl et al., 2023) | ✓ | 25.246 | 0.771 | 0.205 | 21.520 | 0.605 | 0.336 | 27.410 | 0.868 | 0.103 | 26.550 | 0.775 | 0.210 | 22.490 | 0.638 | 0.317 |
| 3DGS(re-run) (Kerbl et al., 2023) | ✓ | 25.195 | 0.764 | 0.211 | 21.507 | 0.602 | 0.339 | 27.325 | 0.863 | 0.108 | 26.689 | 0.771 | 0.216 | 22.472 | 0.632 | 0.328 |
| Mip-Splatting (Yu et al., 2023) | ✓ | 25.250 | 0.765 | 0.243 | 21.600 | 0.605 | 0.371 | 27.470 | 0.869 | 0.124 | 26.640 | 0.774 | 0.251 | 22.650 | 0.633 | 0.381 |
| 2DGS (Huang et al., 2024) | ✓ | 24.770 | 0.733 | 0.302 | 21.140 | 0.572 | 0.403 | 26.690 | 0.843 | 0.166 | 26.200 | 0.758 | 0.299 | 22.360 | 0.616 | 0.433 |
| 3DGS (Kerbl et al., 2023) | ✗ | 23.781 | 0.652 | 0.333 | 20.450 | 0.539 | 0.384 | 26.417 | 0.834 | 0.140 | 23.067 | 0.667 | 0.303 | 21.456 | 0.593 | 0.385 |
| **RAIN-GS (Ours)** | ✗ | 25.373 | 0.750 | 0.244 | 22.118 | 0.632 | 0.315 | 27.277 | 0.863 | 0.110 | 27.029 | 0.783 | 0.207 | 22.887 | 0.647 | 0.328 |

| Method | SfM points | Mip-NeRF360 Indoor Scene | | | | | | | | | | | | Mip-NeRF360 Average | | |
| | | room | | | counter | | | kitchen | | | bonsai | | | | | |
| | | PSNR↑ | SSIM↑ | LPIPS↓ | PSNR↑ | SSIM↑ | LPIPS↓ | PSNR↑ | SSIM↑ | LPIPS↓ | PSNR↑ | SSIM ↑ | LPIPS↓ | PSNR↑ | SSIM ↑ | LPIPS↓ |
|---|---|---|---|---|---|---|---|---|---|---|---|---|---|---|---|---|
| Plenoxels Yu et al. (2021) | ✗ | 27.594 | 0.842 | 0.419 | 23.624 | 0.759 | 0.441 | 23.420 | 0.648 | 0.447 | 24.669 | 0.814 | 0.398 | 23.080 | 0.625 | 0.462 |
| INGP-Base Müller et al. (2022) | ✗ | 29.269 | 0.855 | 0.301 | 26.439 | 0.798 | 0.342 | 28.548 | 0.818 | 0.254 | 30.337 | 0.890 | 0.227 | 25.302 | 0.671 | 0.371 |
| INGP-BigMüller et al. (2022) | ✗ | 29.690 | 0.871 | 0.261 | 26.691 | 0.817 | 0.306 | 29.479 | 0.858 | 0.195 | 30.685 | 0.906 | 0.205 | 25.586 | 0.699 | 0.331 |
| 3DGS (Kerbl et al., 2023) | ✓ | 30.632 | 0.914 | 0.220 | 28.700 | 0.905 | 0.204 | 30.317 | 0.922 | 0.129 | 31.980 | 0.938 | 0.205 | 27.205 | 0.815 | 0.214 |
| 3DGS(re-run) (Kerbl et al., 2023) | ✓ | 31.538 | 0.918 | 0.224 | 28.989 | 0.906 | 0.204 | 31.181 | 0.925 | 0.129 | 32.266 | 0.941 | 0.209 | 27.462 | 0.814 | 0.219 |
| 2DGS (Huang et al., 2024) | ✓ | 30.370 | 0.906 | 0.317 | 28.100 | 0.892 | 0.292 | 30.410 | 0.916 | 0.179 | 31.300 | 0.931 | 0.280 | 26.810 | 0.796 | 0.297 |
| Mip-Splatting (Yu et al., 2023) | ✓ | 31.540 | 0.918 | 0.286 | 29.040 | 0.907 | 0.258 | 31.250 | 0.926 | 0.155 | 31.960 | 0.941 | 0.254 | 27.490 | 0.815 | 0.258 |
| 3DGS (Kerbl et al., 2023) | ✗ | 29.987 | 0.893 | 0.267 | 27.963 | 0.874 | 0.253 | 30.353 | 0.914 | 0.143 | 29.562 | 0.905 | 0.249 | 25.893 | 0.764 | 0.273 |
| **RAIN-GS (Ours)** | ✗ | 30.866 | 0.916 | 0.218 | 28.681 | 0.905 | 0.195 | 31.416 | 0.926 | 0.125 | 31.610 | 0.940 | 0.188 | 27.473 | 0.818 | 0.215 |

| Methods | SfM points | Tanks&Temples | | | | | | Deep Blending | | | | | |
| | | Truck | | | Train | | | DrJohnson | | | Playroom | | |
| | | PSNR↑ | SSIM↑ | LPIPS↓ | PSNR↑ | SSIM↑ | LPIPS↓ | PSNR↑ | SSIM↑ | LPIPS↓ | PSNR↑ | SSIM ↑ | LPIPS↓ |
|---|---|---|---|---|---|---|---|---|---|---|---|---|---|
| Plenoxels Yu et al. (2021) | ✗ | 23.221 | 0.774 | 0.335 | 18.927 | 0.663 | 0.422 | 23.142 | 0.787 | 0.521 | 22.980 | 0.802 | 0.465 |
| INGP-Base Müller et al. (2022) | ✗ | 23.260 | 0.779 | 0.274 | 20.170 | 0.666 | 0.386 | 27.750 | 0.839 | 0.381 | 19.483 | 0.754 | 0.465 |
| INGP-Big Müller et al. (2022) | ✗ | 23.383 | 0.800 | 0.249 | 20.456 | 0.689 | 0.360 | 28.257 | 0.854 | 0.352 | 21.665 | 0.779 | 0.428 |
| 3DGS (Kerbl et al., 2023) | ✓ | 25.187 | 0.879 | 0.148 | 21.097 | 0.802 | 0.218 | 28.766 | 0.899 | 0.244 | 30.044 | 0.906 | 0.241 |
| 3DGS(re-run) (Kerbl et al., 2023) | ✓ | 25.344 | 0.878 | 0.149 | 21.965 | 0.811 | 0.209 | 29.098 | 0.898 | 0.247 | 29.865 | 0.901 | 0.246 |
| Mip-Splatting (Yu et al., 2023) | ✓ | 24.360 | 0.857 | 0.108 | 21.820 | 0.795 | 0.172 | 28.804 | 0.898 | 0.242 | 30.118 | 0.908 | 0.235 |
| 2DGS (Huang et al., 2024) | ✓ | 23.830 | 0.843 | 0.123 | 16.410 | 0.583 | 0.529 | 28.894 | 0.897 | 0.259 | 30.009 | 0.900 | 0.260 |
| 3DGS (Kerbl et al., 2023) | ✗ | 22.685 | 0.821 | 0.201 | 21.039 | 0.768 | 0.254 | 28.874 | 0.892 | 0.262 | 30.172 | 0.901 | 0.253 |
| **RAIN-GS (Ours)** | ✗ | 24.816 | 0.865 | 0.169 | 21.436 | 0.786 | 0.244 | 28.675 | 0.896 | 0.260 | 30.165 | 0.903 | 0.250 |

Table 3: **Quantitative comparison on Mip-NeRF360, Tanks&Temples and Deep Blending datasets**. We compare our method with previous approaches trained from either SfM or random point clouds. Ours trained from random point clouds show competitive performance with previous methods that utilize SfM initializations.

| Methods | COLMAP | scene12 | | | scene20 | | | scene30 | | | scene46 | | | scene57 | | | Avg. PSNR | Avg. Time |
| | | PSNR↑ | SSIM↑ | LPIPS↓ | PSNR↑ | SSIM↑ | LPIPS↓ | PSNR↑ | SSIM↑ | LPIPS↓ | PSNR↑ | SSIM↑ | LPIPS↓ | PSNR↑ | SSIM↑ | LPIPS↓ | | |
|---|---|---|---|---|---|---|---|---|---|---|---|---|---|---|---|---|---|---|
| 3DGS Kerbl et al. (2023) | ✓ | 31.647 | 0.944 | 0.090 | 32.454 | 0.964 | 0.061 | 36.555 | 0.967 | 0.105 | 29.089 | 0.923 | 0.105 | 21.282 | 0.741 | 0.283 | 30.205 | (6m 24s)† + 6m 82s |
| 3DGS(Random) Kerbl et al. (2023) | ✗ | 18.750 | 0.689 | 0.326 | 13.183 | 0.427 | 0.493 | 30.113 | 0.930 | 0.128 | 24.298 | 0.841 | 0.195 | 19.946 | 0.571 | 0.303 | 21.258 | 9m 01s |
| **RAIN-GS (Ours)** | ✗ | 32.822 | 0.953 | 0.075 | 34.730 | 0.972 | 0.051 | 36.767 | 0.967 | 0.080 | 32.298 | 0.953 | 0.074 | 24.095 | 0.838 | 0.213 | 32.142 | 6m 74s |

Table 4: **Quantitative comparison on RealEstate-10K dataset**. We compare our method with 3DGS trained from either SfM or random point cloud on randomly sampled scenes. Ours achieve better performance and time when compared to 3DGS, even without the preprocessing of COLMAP (Schonberger & Frahm, 2016). † refers to the time spent running COLMAP to obtain the SfM point clouds.

When comparing the results of Ours and 3DGS from random point clouds, both point clouds are initialized in the same bound following 3DGS. We follow the same protocol for all datasets.

## 6.4 QUANTITATIVE AND QUALITATIVE COMPARISON

**Quantitative comparisons of image quality.** To assess the image quality, we report the PSNR, LPIPS, and SSIM metrics of the synthesized images. We show the quantitative comparison on Mip-NeRF360, Tanks&Temples, and Deep Blending dataset on Table 3 and the quantitative comparison on RealEstate-10K dataset on Table 4. In all datasets, our method shows competitive or even better performance even when trained with random point clouds outperforming other methods trained with point clouds achieved from SfM, demonstrating the effectiveness of our strategy.

**Quantitative comparisons of training time.** In Table 4, we compare the execution times of COLMAP+3DGS, 3DGS (Random), and our method in a setting where initial SfM point cloud is not available. Our method achieves the best performance in the shortest time, whereas 3DGS requires running COLMAP to obtain the point cloud, which takes almost as much time as the training itself in order to achieve good performance. This demonstrates that, in scenarios where a high-quality point cloud is unavailable, our method can deliver superior performance in significantly less time.

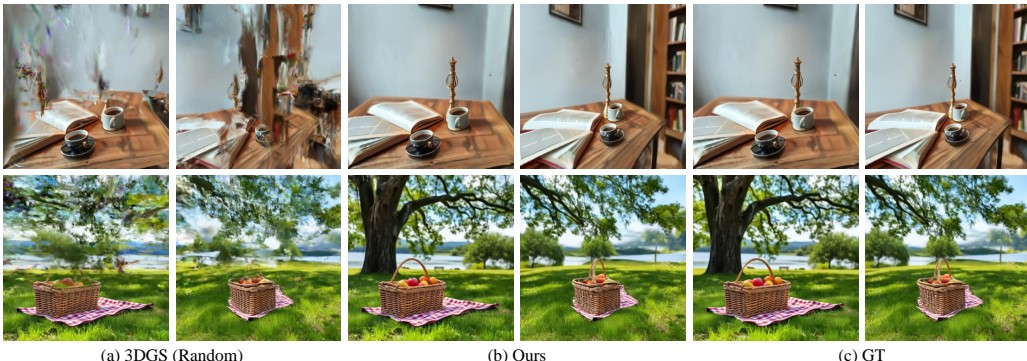

(a) 3DGS (Random)  (b) Ours  (c) GT

Figure 5: **Qualitative results of 3DGS (Random) and Ours trained with generated images using DimensionX (Sun et al., 2024).**

**Qualitative comparisons.** To qualitatively demonstrate the effectiveness of our method, we visually compare the image quality of each method across various datasets. We present the results in Figure 3 and Figure 4. The red bounding boxes illustrate the regions that 3DGS using random point clouds experiences issues due to under-reconstruction. In contrast, our method, despite its simplicity, exhibits its effectiveness in addressing these challenges showing the results on par with the one of 3DGS using SfM point clouds. Additional results can be found in Section C in Appendix.

**Ablation studies.** In Table 5, we validate the effectiveness of each component in our method trained in the Mip-NeRF360 dataset. For the ablation of our SLV initialization, we directly compare the performance of the original random initialization method $N = 100K$. As SLV becomes similar to the original random initialization setting due to the splitting method as

| Low-pass filter | Init. | ABE-Split | PSNR↑ | SSIM↑ | LPIPS↓ |
|---|---|---|---|---|---|
| Constant | $N = 100K$ | ✗ | 25.893 | 0.764 | 0.273 |
| Constant | $N = 100K$ | ✓ | 26.970 | 0.805 | 0.227 |
| Constant | SLV | ✗ | 25.815 | 0.759 | 0.280 |
| Constant | SLV | ✓ | 26.395 | 0.785 | 0.231 |
| Ours | SLV | ✗ | 26.288 | 0.769 | 0.273 |
| **Ours** | **SLV** | ✓ | **27.473** | **0.818** | **0.215** |

Table 5: **Ablation on core components**

mentioned in Section 5.2, using SLV alone does not show any improvements over the original random initialization. However, when combined with the low-pass filter and ABE-Split algorithm, SLV initialization shows the best performance verifying our design choice. More detailed ablation studies can be found in Section A.3 of the Appendix.

**Training with generated images.** Our method can be effectively utilized when training with generated images. We train 3DGS from random point clouds with the generated images from DimensionX (Sun et al., 2024), with the camera poses given as condition. Both qualitative results in Figure 5 and quantitative

| Methods | Scene 1 | | | Scene 2 | | |
|---|---|---|---|---|---|---|
| | PSNR↑ | SSIM↑ | LPIPS↓ | PSNR↑ | SSIM↑ | LPIPS↓ |
| 3DGS(Random) | 12.278 | 0.712 | 0.413 | 11.173 | 0.404 | 0.509 |
| **Ours** | **32.751** | **0.951** | **0.129** | **25.262** | **0.789** | **0.216** |

Table 6: **Quantitative comparison on generated images using DimensionX (Sun et al., 2024).**

results in Table 6 verify the effectiveness of our method. Note that SfM algorithms (Schonberger & Frahm, 2016) fail to converge in these images failing to provide initial point clouds.

# 7 CONCLUSION

In this work, we introduced **RAIN-GS**, a novel strategy to address the limitations of 3D Gaussian Splatting (3DGS), particularly its reliance on accurate initial point clouds and the limited transportability of Gaussians. By leveraging Sparse-Large-Variance (SLV) initialization, progressive Gaussian low-pass filtering, and the Adaptive Bound-Expanding (ABE) split algorithm, RAIN-GS effectively mitigates under-reconstruction issues, enabling Gaussians to explore the scene more globally and improve reconstruction quality. Our extensive experiments demonstrate that RAIN-GS achieves competitive or superior results even when using random initializations, significantly reducing the dependence on high-quality point clouds and making 3DGS a more robust solution for novel view synthesis. We believe that our RAIN-GS can broaden the applicability of 3DGS in real-world scenarios where obtaining accurate initial point clouds may not be feasible.

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

## A    DETAILS OF ANALYSIS

### A.1    ANALYSIS OF VARIOUS POINT CLOUD INITIALIZATIONS

| | Outdoor Scene | | | | | | | | | | | | | | |
|---|---|---|---|---|---|---|---|---|---|---|---|---|---|---|---|
| Initialization | bicycle | | | flowers | | | garden | | | stump | | | treehill | | |
| | PSNR↑ | SSIM↑ | LPIPS↓ | PSNR↑ | SSIM↑ | LPIPS↓ | PSNR↑ | SSIM↑ | LPIPS↓ | PSNR↑ | SSIM↑ | LPIPS↓ | PSNR↑ | SSIM↑ | LPIPS↓ |
| SfM | 25.195 | 0.764 | 0.211 | 21.507 | 0.602 | 0.339 | 27.325 | 0.863 | 0.108 | 26.689 | 0.771 | 0.216 | 22.472 | 0.632 | 0.328 |
| SfM + ε | 24.836 | 0.729 | 0.267 | 21.190 | 0.575 | 0.368 | 27.043 | 0.854 | 0.125 | 26.479 | 0.762 | 0.233 | 22.455 | 0.625 | 0.356 |
| SfM+constant | 23.619 | 0.625 | 0.358 | 21.139 | 0.569 | 0.364 | 25.663 | 0.809 | 0.163 | 23.382 | 0.641 | 0.335 | 21.989 | 0.593 | 0.380 |

| | Indoor Scene | | | | | | | | | | | | Average | | |
|---|---|---|---|---|---|---|---|---|---|---|---|---|---|---|---|
| Initialization | room | | | counter | | | kitchen | | | bonsai | | | | | |
| | PSNR↑ | SSIM↑ | LPIPS↓ | PSNR↑ | SSIM↑ | LPIPS↓ | PSNR↑ | SSIM↑ | LPIPS↓ | PSNR↑ | SSIM↑ | LPIPS↓ | PSNR↑ | SSIM↑ | LPIPS↓ |
| SfM | 31.538 | 0.918 | 0.224 | 28.989 | 0.906 | 0.204 | 31.181 | 0.925 | 0.129 | 32.266 | 0.941 | 0.209 | 27.462 | 0.814 | 0.219 |
| SfM + ε | 31.038 | 0.907 | 0.249 | 28.211 | 0.888 | 0.233 | 29.863 | 0.915 | 0.141 | 31.922 | 0.935 | 0.219 | 27.004 | 0.799 | 0.243 |
| SfM+constant | 30.790 | 0.899 | 0.264 | 28.041 | 0.875 | 0.250 | 30.203 | 0.914 | 0.145 | 31.006 | 0.924 | 0.230 | 26.204 | 0.761 | 0.277 |

Table 7: **Quantitative comparison on Mip-NeRF360 dataset in noisy initial SfM point cloud settings**. We compare 3DGS method with different noisy inital SfM point cloud. We report PSNR, SSIM, LPIPS.

In Table 1, we present detailed results to further investigate the ability of the 3DGS optimization scheme to transport Gaussians to the correct 3DGS locations on Mip-NeRF360 datasets. Here, we conduct this experiments by adding random noise $\epsilon \sim \mathcal{N}(0, 0.5)$ and constant systematic noise, whose value equals 2, to the initial SfM points. The results shown in Table 7 prove that 3DGS strongly depends on the initial point. Figure 6 shows initial SfM points and points with noise $\epsilon$. Even with the small amount of noise, 3DGS fails to move to the correct position.

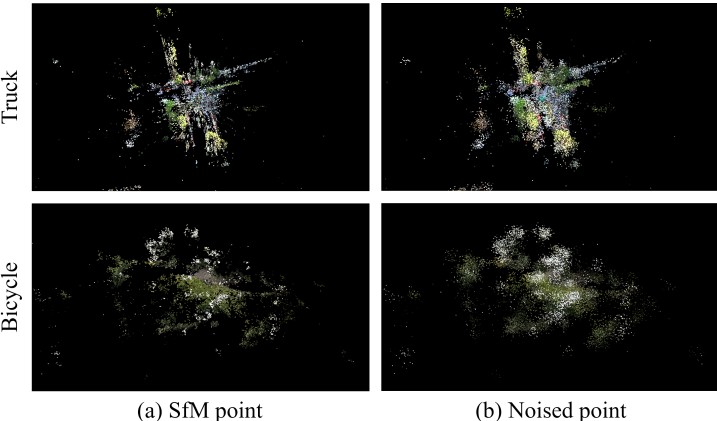

(a) SfM point                    (b) Noised point

Figure 6: **Visualization of SfM points and points with noise $\epsilon$.**

### A.2    ANALYSIS OF MOVEMENT OF EACH GAUSSIAN

In our analysis of Gaussian movement, we have carefully accounted for the complex dynamics introduced by cloning and splitting processes. We track these Gaussians throughout the optimization process, maintaining their original identifiers as they undergo cloning or splitting events. This approach allows us to trace the lineage of each Gaussian from its initial state to its final position.

Experiment details are as follows : Assume that we have 10 initial Gaussians, saving the coordinates of each to track their movement. To distinguish them, each Gaussian is assigned a label ranging from 1 to 10. Throughout the process, Gaussians undergo cloning and splitting. When a Gaussian is cloned, the new Gaussian retains the original label. Similarly, when a Gaussian undergoes splitting, both resulting Gaussians are assigned the same label.

Since new Gaussians are generated solely through cloning and splitting, by the end of the optimization, we have N Gaussians, each still labeled within the original range of 1 to 10. To calculate the

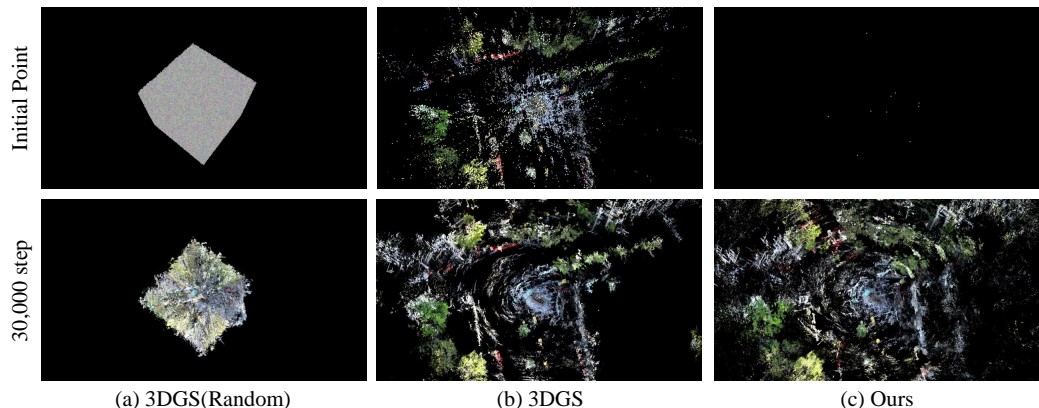

(a) 3DGS(Random)  (b) 3DGS  (c) Ours

Figure 7: **Displacements of Gaussians from initial positions.**

overall movement, we determine the displacement of each Gaussian by measuring the Euclidean distance between the post-optimization coordinates of the N Gaussians and the initial coordinates of the 10 original Gaussians.

To observe the movement of the each Gaussian, we measure how far the Gaussian moves from its initial position during the training. An additional parameter is incorporated to record the initial position, ensuring that even when Gaussians are split or cloned, the initial position parameters are retained. Then, the movement is calculated as difference between the final position of each Gaussian after training and its respective initial position.

We conduct analysis on "Truck" scene of Tanks&Temples dataset, comparing the settings of 3DGS with SfM point initialization, 3DGS with random initialization, and our method. The mean, standard deviation, and the top 1% values of the movements are shown in Table 2. Additionally, Figure 7 shows the overall scene from the same camera viewpoint for each experiment to observe the differences in distribution of overall Gaussians between the beginning and 30,000 steps. In case such as 3DGS with SfM initialization and random initialization, the positions of the Gaussians does not change significantly. However, our method shows substantial changes in comparison.

| Scene | $x$ | $y$ | $z$ |
|---|---|---|---|
| treehill | 156.24 | 62.91 | 155.93 |
| flowers | 89.89 | 35.69 | 80.28 |
| stump | 209.00 | 156.39 | 219.48 |
| counter | 25.84 | 24.39 | 26.53 |
| garden | 100.88 | 41.41 | 59.47 |
| bicycle | 108.81 | 43.51 | 138.92 |
| kitchen | 49.25 | 44.02 | 64.99 |
| room | 45.80 | 34.08 | 59.80 |
| bonsai | 40.61 | 35.73 | 48.83 |
| Average | 91.81 | 53.13 | 94.91 |

Table 8: **Bounds for each scene in the MipNeRF360 dataset.** Each value is calculated by the bound of SfM point clouds and represents the width in the corresponding direction.

### A.3 ABLATION STUDY OF PROGRESSIVE LOW-PASS FILTER CONTROL

To demonstrate the effectiveness of our progressive Gaussian low-pass filter control strategy, we employ three different decreasing functions of convex, linear, and concave to control the Gaussian low-pass filter value $s$. Different from our strategy, where the value $s$ is defined adaptively by

image height, width, and the number of Gaussians $N$ at each time step, the remaining functions are manually defined to achieve $s = 300$ at step 0 and $s = 0.3$ at about 3,000 steps across all scenes. The intuition behind this design is based on our analysis that our adaptive Gaussian low-pass filter value reaches 0.3 between 2,000-3,000 steps. Also, we empirically find that the initial Gaussian low-pass filter value $s > 300$ offers no significant improvement, only making the overall computation inefficient. Based on these findings, we define the max value of the Gaussian low-pass filter as $s = 300$.

For the convex function, we use the following formula for $s$ scheduling:

$$s = \max(7^{-\frac{x}{1000}} * 300, 0.3).$$

(6)

For the linear function, we use the following formula for $s$ scheduling:

$$s = \max(300 - 0.0997084x, 0.3).$$

(7)

For the concave function, we use the following formula for $s$ scheduling:

$$s = \max(300 * (1 + 7^{-3} - 7^{\frac{x-3000}{1000}}), 0.3).$$

(8)

The illustration of different Gaussian low-pass value formulas is shown in Figure 8 and Figure 9 where our formula is adaptively defined, showing different functions for each scene.

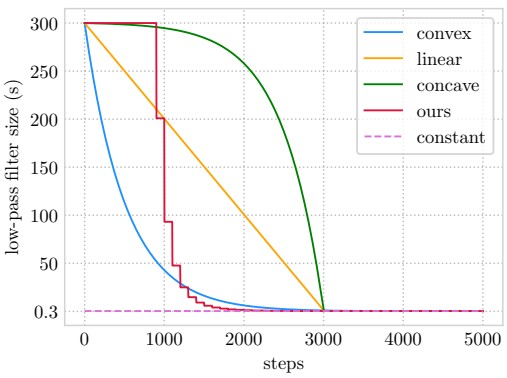

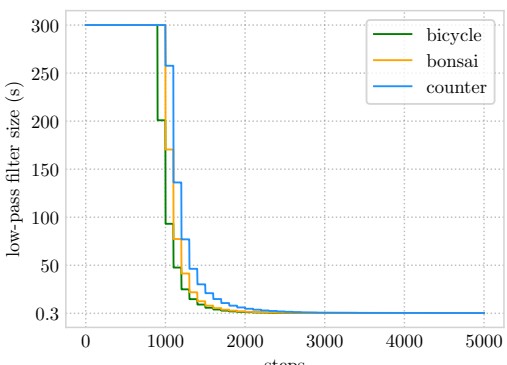

Figure 8: **Illustration of different Gaussian low-pass filter value formulas.**

Figure 9: **Illustration of our Gaussian low-pass filter value formula.**

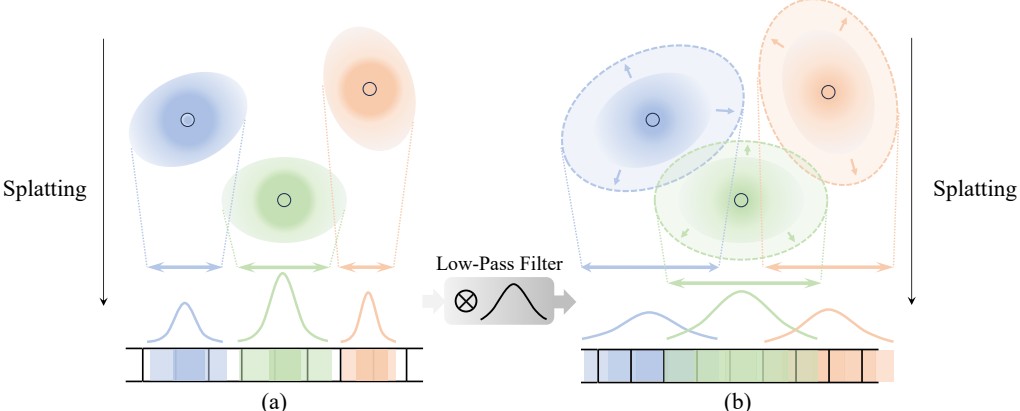

Figure 10: **Visualization of low-pass filter.** This figure shows the visualization of the effect of the low-pass filter. As shown in (b), the convolution of the splatted 2D Gaussian with the low-pass filter expands the area the Gaussian is splatted onto, resulting in the Gaussians affecting larger areas than naïve splatting as shown in (a).

## B PROOF

### B.1 PROOF ON RADIUS OF A GAUSSIAN CONVOLVED WITH A LOW-PASS FILTER

As mentioned in Section 5.2 of our main paper, the 3D Gaussians $G_i$ is projected to 2D Gaussians $G'_i$ in the screen space as follows:

$$G'_i(x) = e^{-\frac{1}{2}(x-\mu'_i)^T \Sigma'^{-1}_i (x-\mu'_i)}. \tag{9}$$

To ensure the 2D Gaussian $G'_i$ to cover at least one pixel, 3DGS adds a small value $s$ to the diagonal elements of the 2D covariance $\Sigma'_i$ as follows:

$$G'_i(x) = e^{-\frac{1}{2}(x-\mu'_i)^T (\Sigma'_i + sI)^{-1} (x-\mu'_i)}, \tag{10}$$

where $I$ is the $2 \times 2$ identity matrix. This process can be understood as the convolution between the 2D Gaussian $G'_i$ and the Gaussian low-pass filter $h$ (mean $\mu = 0$ and variance $\sigma^2 = s = 0.3$) of $G'_i \otimes h$. This is due to the nature of Gaussians where the convolution of Gaussians with the variance matrices $V$ and $Z$ results in a Gaussian with the variance matrix $V + Z$ as follows:

$$G_1(x) = e^{-\frac{1}{2}(x-\mu_i)^T V^{-1} (x-\mu_i)} \quad G_2(x) = e^{-\frac{1}{2}(x-\mu_i)^T Z^{-1} (x-\mu_i)}, \tag{11}$$

$$(G_1 \otimes G_2)(x) = e^{-\frac{1}{2}(x-\mu_i)^T (V+Z)^{-1} (x-\mu_i)}. \tag{12}$$

Following the convolution process, 3DGS estimates the projected 2D Gaussian's area to identify its corresponding screen tiles. This is done by calculating $k$ times the square root of the larger eigenvalue of $(\Sigma'_i + sI)$, which represents the radius of the approximated circle, and $k$ is the hyperparameter that determines the confidence interval of the 2D Gaussian. Figure 10 illustrates the low-pass filter's effect, where the projected Gaussian is splatted to wider areas in (b) compared to (a).

### B.2 PROOF ON PROGRESSIVE LOW-PASS FILTER SIZE

In Section 5.2 of our main paper, we define the value $s$ for our progressive Gaussian low-pass filter control based on the fact that the area of the projected 2D Gaussians is at least $9\pi s$. As the area of the projected 2D Gaussian is defined as the circle whose radius is $k$ times the square root of the larger eigenvalue of $(\Sigma'_i + sI)$, we have to first calculate the eigenvalues of $(\Sigma'_i + sI)$. If we define the eigenvalues of $\Sigma_i$ as $\lambda_{i1}, \lambda_{i2}$, since the eigenvalue of $sI$ is $s$, the eigenvalues of $(\Sigma'_i + sI)$ can

be defined as $\lambda_{i1} + s, \lambda_{i2} + s$. This leads to the following proof:

$$r = k \cdot \sqrt{\max(\lambda_{i1}, \lambda_{i2}) + s},$$
$$r \geq k \cdot \sqrt{s}, \tag{13}$$
$$\pi r^2 \geq k^2 \pi s,$$

where $k$ is the hyperparameter that defines the confidence interval of the Gaussian. We follow the original implementation of 3DGS as $k = 3$ which gives the $99.73\%$ confidence interval. Using the value $k = 3$ leads to the proof of the area of each Gaussian being at least $9\pi s$.

# C  ADDITIONAL QUALITATIVE RESULTS

We show additional qualitative results in Figure 11.

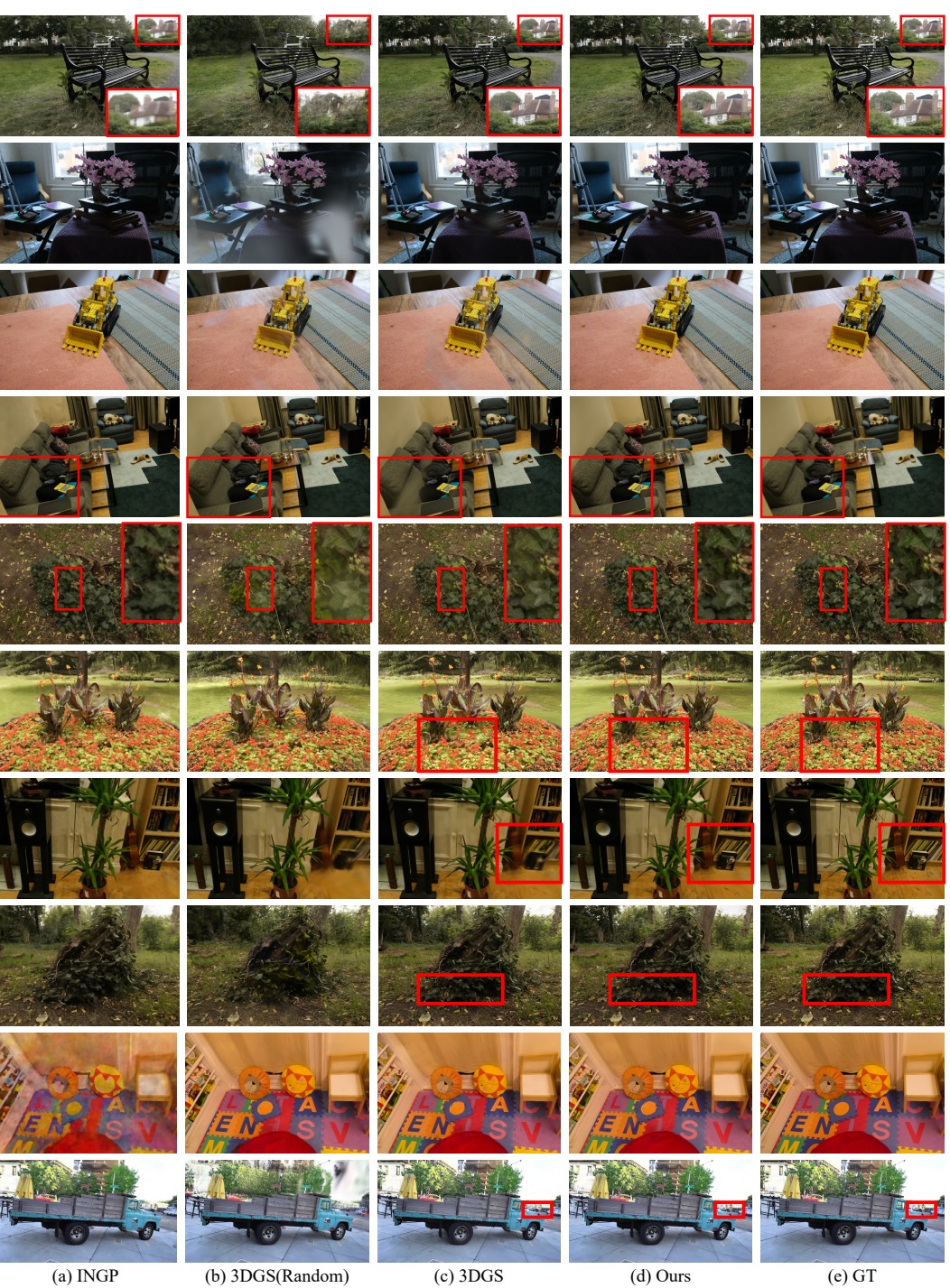

|  (a) INGP  |  (b) 3DGS(Random)  |  (c) 3DGS  |  (d) Ours  |  (e) GT |

Figure 11: **Additional qualitative results.**

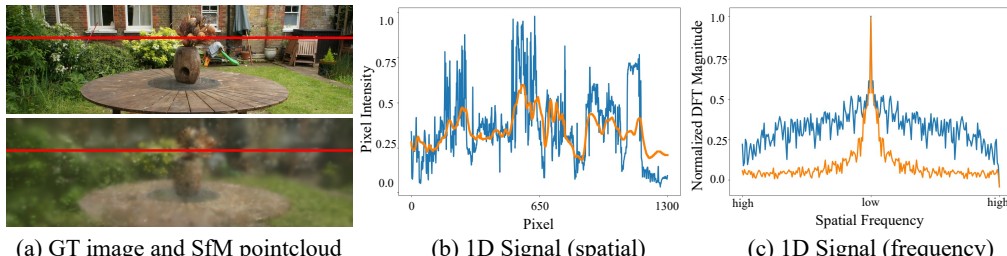

(a) GT image and SfM pointcloud     (b) 1D Signal (spatial)     (c) 1D Signal (frequency)

Figure 12: **Analysis of SfM initialization in 3DGS.** (a) The top shows the GT image, and the bottom is the rendered image by 3DGS after only 10 steps with SfM initialization. We can observe that the rendered image is already coarsely-close to GT image. We randomly sample a horizontal line from the image marked in red. (b) The pixel intensity along this line are shown, with the GT indicated in blue and the rendered image in orange. (c) This graph visualizes the magnitude of the frequency components of (b). Since frequencies further from the middle of the x-axis represent high-frequency components, we observe that SfM provides *coarse* approximation of the true distribution.

## D   ADDITIONAL ANALYSIS

Although point cloud can be noisy or unavailable in real-world scenarios (Bian et al., 2023; Zhang et al., 2022), 3DGS shows large performance drops depending on the accuracy of the initial point cloud (Kerbl et al., 2023). To understand the large performance gap of 3DGS, we conducted an in-depth analysis of the original 3DGS optimization scheme in Section 4 of the main paper. In this section, we further explore additional benefits from accurate initialization. Specifically, we analyze SfM initialization in the frequency domain. Our analyses reveal two important characteristics of the optimization scheme of 3DGS: **1)** the optimization scheme of 3DGS struggles to transport Gaussians from their initialized locations and **2)** the coarse structure information (low-frequency components) provided by the accurate initialization enables the adaptive density control method of 3DGS to robustly model the remaining fine details of the scene in a coarse-to-fine manner.

**3DGS lacks the ability to transport Gaussians.**     To represent and learn the scene with explicit 3D Gaussians, 3DGS first initializes the Gaussians $G_i$ in the world space, whose means $\mu_i$ are defined by the initial point cloud. The point cloud can be either achieved from SfM or initialized randomly.

As mentioned in (Charatan et al., 2023), the process of fitting a 3DGS model is similar to fitting a Gaussian Mixture Model (GMM), which is well-known for being non-convex and generally solved with the Expectation-Maximization (EM) algorithm (Dempster et al., 1977). They further note that, similar to the EM algorithm, training 3DGS from randomly initialized point cloud becomes prone to falling into local minima due to two main reasons. **1)** The Gaussians can only receive gradients close to their means, mostly from the range not exceeding the distance of a few standard deviations, and **2)** there is no existing path for the Gaussians that will decrease the loss monotonically.

Although (Charatan et al., 2023) only analyzes the case of starting from random initialization, we verify that Gaussians can also easily fall into local minima when SfM-initialized point cloud become noisy. As shown in Table 1, we find that adding a small constant noise or adding a small noise $\epsilon$ sampled from a normal distribution ($\epsilon \sim \mathcal{N}(0, 0.5)$) to the SfM-initialized point cloud, leads to large performance drops. Based on these observations, we hypothesize that the optimization scheme of 3DGS lacks the ability to correct or move the positions of the Gaussians. We empirically verify our hypothesis by calculating the average distance each Gaussian traversed after optimization, as shown in Table 2. It can be seen that the average distance each Gaussian moved is close to zero, indicating that the optimization scheme of 3DGS lacks the ability to move Gaussians, which can lead to the failure of capturing objects located far from the initial positions of the Gaussians. This emphasizes the need for a strategy that can enable the Gaussians to transport further from their initialized locations, in order to successfully train 3DGS from sub-optimal initializations.

**Accurate initialization guides 3DGS to learn in a coarse-to-fine manner.**     To further investigate the benefits of SfM initialization, we analyze the rendered images in the frequency domain using Fourier transform (Nussbaumer & Nussbaumer, 1982). As shown in Figure 12, the analysis in the frequency domain demonstrates that SfM initialization provides a coarse approximation of the target distribution.

As the goal of novel view synthesis is to understand the 3D distribution of the scene, it is necessary to model both low- and high-frequency components of the true distribution. However, prior NeRF frameworks (Lin et al., 2021; Park et al., 2021; Yang et al., 2023) argue that NeRF is prone to overfitting and naïve optimization leads to over-fast convergence of high-frequency components, expressed with high-frequency artifacts in the rendered image. To circumvent this problem, they adopt a coarse-to-fine learning strategy, which regularizes NeRF to learn the low-frequency components first. Similarly, prior works (Eckart et al., 2016; Hertz et al., 2020) utilizing GMMs for the task of point cloud registration or generation also mention that naïve fitting of GMMs can result in converging to local minima. In order to robustly train GMMs, they also adopt a coarse-to-fine strategy, implemented by starting with a small number of Gaussians and recursively increasing the number of total Gaussians. In both NeRFs and GMMs, coarse-to-fine strategy guides the network to learn more robustly, leading to better performance.

In this perspective, starting the optimization of 3DGS from SfM-initialized point cloud can be understood as benefitting from a similar coarse-to-fine process, where SfM provides the low-frequency components (Figure 12), and the adaptive density control method of 3DGS adds the Gaussians to learn the remaining high-frequency details. Based on our observations, the success of 3DGS from accurate initialization can be attributed to the low-frequency components guiding the overall training process, preventing the Gaussians from falling into local minima. This highlights the need for a strategy that can prioritize the learning of the low-frequency components even from sub-optimal initializations, which will then be used to guide the remaining optimization process of 3DGS.

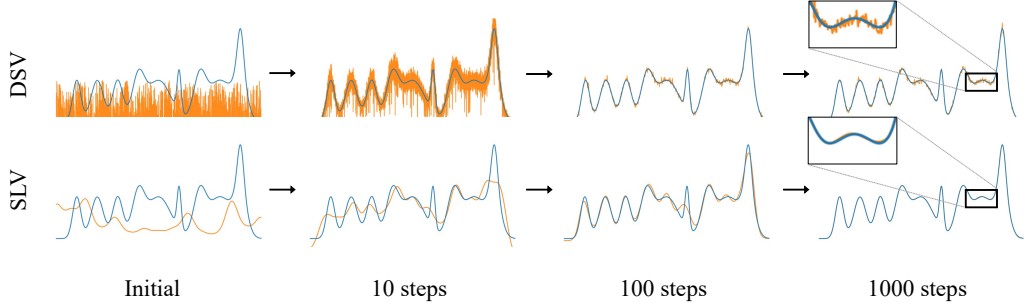

Figure 13: **Toy experiment to analyze different initialization methods.** This figure visualizes the result of our toy experiment predicting the target distribution using a collection of 1D Gaussians, starting from different initialization methods.

# E ADDITIONAL INTERPRETATION ABOUT SPARSE-LARGE-VARIANCE (SLV) INITIALIZATION

Drawing inspiration from GMMs (Eckart et al., 2016; Nichol et al., 2022), which gradually increase the number of Gaussians to accurately model target point cloud, we observe that the adaptive density control of 3DGS can be viewed as a similar process. Through cloning and splitting operations, 3DGS generally increases the number of Gaussians to find the adequate number of Gaussians required to represent the scene. Based on our findings, we hypothesize that initializing 3DGS with a sparse set of Gaussians will prioritize the learning of low-frequency components, akin to the progressive refinement approach employed by GMMs. This sparse initialization strategy is expected to capture the overall structure of the target point cloud in the early stages of the optimization process, with finer details being added as the number of Gaussians increases.

To verify our hypothesis, we conduct a toy experiment in a simplified 1D regression task. Following the original 3DGS which can be interpreted as the learning process of a 3D target distribution with multiple Gaussians, we use $N$ Gaussians each with learnable means, variances, and weights, which are then blended to model a 1D target signal. Specifically, we follow the initialization methods of 3DGS (Kerbl et al., 2023), where the means are initialized randomly and the variances are initialized based on the distances of the three nearest neighbors. As a result, sparse initialization of Gaussians leads to a larger initial covariance (SLV) and dense initialization leads to a smaller covariance (DSV). To verify our hypothesis that learning with sparse Gaussians will prioritize the learning of low-frequency components, we conduct our toy experiment using $N = 15$ and $N = 1000$ for the SLV and DSV initialization respectively. Note that our 1D toy experiment without the adaptive density control method of 3DGS provides a controlled environment isolating the effects of initialization.

As shown in Figure 13, SLV initialization prioritizes the learning of low-frequency components compared to DSV initialization verifying our hypothesis. After 1,000 steps, SLV also shows a better prediction of the target distribution. Similar results can be observed when SLV is applied to 3DGS, as lowering the number of initial Gaussians $N$ in randomly initialized settings significantly improves performance. Following the random initialization method of (Kerbl et al., 2023), which randomly samples point cloud from a scene extent defined as three times the bounding box of the camera poses, SLV prioritizes the learning of low-frequency components, producing fewer high-frequency artifacts. Surprisingly, SLV becomes more effective even until extremely sparse settings (e.g., as low as $N = 10$), verifying the effectiveness of our novel SLV initialization method.

# F ADDITIONAL EXPERIMENTS

## F.1 ABLATION ON CORE COMPONENTS

In Table 9, we show a more detailed ablation of our core components: SLV initialization, Progressive Gaussian low-pass filtering, and ABE-Split. SLV indicates that we initialize $N = 10$ Gaussians. The ablations verify our choice as leveraging all three of our components yields the best performance.

| Low-pass filter | Init. | ABE-Split | PSNR↑ | SSIM↑ | LPIPS↓ |
|---|---|---|---|---|---|
| Constant | $N = 100K$ | ✗ | 25.893 | 0.764 | 0.273 |
| Constant | $N = 100K$ | ✓ | 26.970 | 0.805 | 0.227 |
| Constant | SLV | ✗ | 25.815 | 0.759 | 0.280 |
| Constant | SLV | ✓ | 26.395 | 0.785 | 0.231 |
| Ours | $N = 100K$ | ✗ | 26.116 | 0.765 | 0.273 |
| Ours | $N = 100K$ | ✓ | 26.982 | 0.808 | 0.226 |
| Ours | SLV | ✗ | 26.288 | 0.769 | 0.273 |
| **Ours** | **SLV** | ✓ | **27.473** | **0.818** | **0.215** |

Table 9: **Ablation on core components.**

## F.2 RAIN-GS WITH OTHER GAUSSIAN SPLATTING METHOD

| Method | SfM points | Mip-NeRF360 Outdoor Scene | | | | | | | | | | | | | | |
|---|---|---|---|---|---|---|---|---|---|---|---|---|---|---|---|---|
| | | bicycle | | | flowers | | | garden | | | stump | | | treehill | | |
| | | PSNR↑ | SSIM↑ | LPIPS↓ | PSNR↑ | SSIM↑ | LPIPS↓ | PSNR↑ | SSIM↑ | LPIPS↓ | PSNR↑ | SSIM↑ | LPIPS↓ | PSNR↑ | SSIM↑ | LPIPS↓ |
| Scaffold-GS (Lu et al., 2024) | ✓ | 24.50 | 0.705 | 0.259 | 21.44 | 0.592 | 0.382 | 27.17 | 0.842 | 0.136 | 26.27 | 0.784 | 0.277 | 23.15 | 0.640 | 0.373 |
| Scaffold-GS (Lu et al., 2024) | ✗ | 23.05 | 0.609 | 0.379 | 19.79 | 0.503 | 0.400 | 26.38 | 0.827 | 0.162 | 22.48 | 0.604 | 0.362 | 21.33 | 0.551 | 0.430 |
| **Scaffold-GS + RAIN-GS (Ours)** | ✗ | 25.32 | 0.738 | 0.268 | 21.82 | 0.619 | 0.312 | 27.75 | 0.866 | 0.106 | 26.45 | 0.757 | 0.238 | 23.06 | 0.641 | 0.324 |

| Method | SfM points | Mip-NeRF360 Indoor Scene | | | | | | | | | | | | Mip-NeRF360 | | |
|---|---|---|---|---|---|---|---|---|---|---|---|---|---|---|---|---|
| | | room | | | counter | | | kitchen | | | bonsai | | | Average | | |
| | | PSNR↑ | SSIM↑ | LPIPS↓ | PSNR↑ | SSIM↑ | LPIPS↓ | PSNR↑ | SSIM↑ | LPIPS↓ | PSNR↑ | SSIM↑ | LPIPS↓ | PSNR↑ | SSIM↑ | LPIPS↓ |
| Scaffold-GS (Lu et al., 2024) | ✓ | 31.93 | 0.925 | 0.275 | 29.34 | 0.910 | 0.256 | 31.30 | 0.928 | 0.156 | 32.70 | 0.946 | 0.249 | 27.53 | 0.808 | 0.263 |
| Scaffold-GS (Lu et al., 2024) | ✗ | 30.65 | 0.896 | 0.264 | 28.15 | 0.875 | 0.253 | 29.43 | 0.890 | 0.181 | 30.66 | 0.918 | 0.241 | 25.77 | 0.741 | 0.297 |
| **Scaffold-GS + RAIN-GS (Ours)** | ✗ | 31.93 | 0.921 | 0.201 | 29.57 | 0.908 | 0.187 | 31.82 | 0.924 | 0.126 | 32.21 | 0.939 | 0.180 | **27.77** | **0.812** | **0.216** |

Table 10: **Quantitative comparison of Scaffold-GS with various initializations on Mip-NeRF360 dataset.**

Our proposed strategy does not involve modifying the model architecture of 3D Gaussian Splatting (3DGS) which enables RAIN-GS to be seamlessly integrated with various 3DGS-based methods in a plug-and-play manner. However, there are also various extensions of 3DGS (Lu et al., 2024) which modify the overall 3DGS optimization algorithm. For these methods, it becomes less straightforward to integrate our method in these approaches. However, instead of directly integrating our method, it is also possible to interpret our method as a coarse-to-fine approach that jointly learns the 3DGS model and an ideal point cloud during the training process. We show that even with the SfM-initialized point clouds being available, the intermediate point clouds generated during the initial stages of RAIN-GS training can serve as superior starting points for other methods.

Specifically, in this section, we show the results of training Scaffold-GS (Lu et al., 2024) with the initial point clouds achieved from our method. Instead of directly training Scaffold-GS with random point clouds, we train RAIN-GS with random point clouds. We find that after training RAIN-GS for 7000 steps, the number of Gaussians is similar to the number of point clouds generated during the SfM pipeline. Therefore, we save the positions of the Gaussians at 7000 steps as point clouds and train Scaffold-GS using these point clouds as initialization. We show the performance of Scaffold-GS trained on Mip-NeRF360 dataset (Barron et al., 2022) in Table 10. The comparison reveals that Scaffold-GS trained with the point clouds obtained from RAIN-GS yields the best performance, even surpassing Scaffold-GS trained with SfM point clouds. This reveals that RAIN-GS can be further utilized to boost the performance of existing 3DGS methods by replacing the initial point clouds and also enabling these methods to be trained even from random point clouds.

| Methods | Mip-NeRF360 Average (w/o camera pose optimization) | | | Mip-NeRF360 Average (w/ camera pose optimization) | | |
|---|---|---|---|---|---|---|
| | PSNR↑ | SSIM↑ | LPIPS↓ | PSNR↑ | SSIM↑ | LPIPS↓ |
| 3DGS (Random) | 16.026 | 0.386 | 0.623 | 18.916 | 0.453 | 0.572 |
| **Ours** | **16.492** | **0.404** | **0.607** | **21.060** | **0.538** | **0.475** |

Table 11: **Quantitative results on noisy camera pose setting.** We evaulate our method with 3DGS (Random) on noisy camera pose setting (Park et al., 2023). Both with and without camera pose optimization, ours achieve better results.

| Method | SfM points | Mip-NeRF360 Outdoor Scene | | | | | | | | | | | | | | |
|---|---|---|---|---|---|---|---|---|---|---|---|---|---|---|---|---|
| | | bicycle | | | flowers | | | garden | | | stump | | | treehill | | |
| | | PSNR↑ | SSIM↑ | LPIPS↓ | PSNR↑ | SSIM↑ | LPIPS↓ | PSNR↑ | SSIM↑ | LPIPS↓ | PSNR↑ | SSIM↑ | LPIPS↓ | PSNR↑ | SSIM↑ | LPIPS↓ |
| RAIN-GS (Resolution) | ✗ | 25.039 | 0.743 | 0.244 | 21.778 | 0.618 | 0.323 | 26.985 | 0.856 | 0.114 | 26.882 | 0.776 | 0.205 | 22.511 | 0.630 | 0.332 |
| **RAIN-GS (Ours)** | ✗ | 25.373 | 0.750 | 0.244 | 22.118 | 0.632 | 0.315 | 27.277 | 0.863 | 0.110 | 27.029 | 0.783 | 0.207 | 22.887 | 0.647 | 0.328 |

| Method | SfM points | Mip-NeRF360 Indoor Scene | | | | | | | | | | | | Mip-NeRF360 Average | | |
|---|---|---|---|---|---|---|---|---|---|---|---|---|---|---|---|---|
| | | room | | | counter | | | kitchen | | | bonsai | | | | | |
| | | PSNR↑ | SSIM↑ | LPIPS↓ | PSNR↑ | SSIM↑ | LPIPS↓ | PSNR↑ | SSIM↑ | LPIPS↓ | PSNR↑ | SSIM↑ | LPIPS↓ | PSNR↑ | SSIM↑ | LPIPS↓ |
| RAIN-GS (Resolution) | ✗ | 30.675 | 0.907 | 0.245 | 28.551 | 0.895 | 0.220 | 31.220 | 0.920 | 0.138 | 31.552 | 0.935 | 0.217 | 27.244 | 0.809 | 0.226 |
| **RAIN-GS (Ours)** | ✗ | 30.866 | 0.916 | 0.218 | 28.681 | 0.905 | 0.195 | 31.416 | 0.926 | 0.125 | 31.610 | 0.940 | 0.188 | **27.473** | **0.818** | **0.215** |

Table 12: **Progressive Low-pass filtering vs. Resolution-based training**.

## F.3 RAIN-GS WITH NOISY CAMERA POSES

When camera poses are obtained solely from sensors (*e.g.,* IMU) or when SfM algorithms fail to provide accurate camera poses, the performance of both NeRF and 3DGS has been shown to degrade significantly (Lin et al., 2021; Fu et al., 2024). To evaluate the robustness of our approach under noisy camera poses, we conduct additional experiments on the Mip-NeRF360 dataset (Barron et al., 2022) with noisy camera poses. Noisy camera poses were generated by following the protocol of CamP (Park et al., 2023), introducing approximately 5 degrees of noise to all poses. As the original rasterizer of 3DGS does not propagate gradients to camera poses, we implemented a custom rasterizer for both 3DGS and our method to enable camera pose correction during training. The average results, both with and without camera pose optimization, are summarized in Table 11. The results show that while noisy camera poses significantly degrade the reconstruction quality, our method consistently outperforms the original random point cloud initialization. The superior performance of our approach can be attributed to the progressive Gaussian low-pass filtering and SLV initialization, which share similarities with the coarse-to-fine training strategies used for handling noisy poses in NeRF (Lin et al., 2021). Combined with our earlier experiments in Table 1, these results demonstrate that our approach can robustly train 3DGS models regardless of whether the initial point cloud or the camera poses are noisy.

## F.4 PROGRESSIVE LOW-PASS FILTERING VS. RESOLUTION-BASED TRAINING

The progressive Gaussian low-pass filtering method is similar to the coarse-to-fine training paradigm, where 3D Gaussians are trained from low-resolution images to high-resolution images. Larger values of the low-pass filter cause Gaussians to cover larger areas, resulting in blurrier images, akin to coarse representations. However, our approach differs from training 3DGS directly from low-resolution images in two significant ways. First, encouraging Gaussians to cover larger areas ensures sufficient gradient propagation during training, enabling effective cloning and splitting of Gaussians and preventing overfitting to highly localized regions. Second, our method uses a consistent high-resolution ground truth for supervision, avoiding the aliasing artifacts that can occur when training with multi-scale images, as noted in Mip-Splatting (Yu et al., 2023). To compare these approaches, we conduct an experiment on the Mip-NeRF360 dataset, replacing our progressive Gaussian low-pass filtering with a coarse-to-fine strategy using low-resolution to high-resolution images, with other strategies remain same. The results, presented in Table 12, show that our method is more robust and achieves better performance across various scenes.

|        | SfM   | Noisy SfM | Random | RAIN-GS (Ours) |
|--------|-------|-----------|--------|----------------|
| Means  | 0.704 | 0.650     | 0.395  | 16.403         |
| Stds   | 2.207 | 0.729     | 0.402  | 14.606         |
| Top 1% | 10.755| 3.646     | 1.923  | 68.919         |

Table 13: **Movement of Gaussians.**

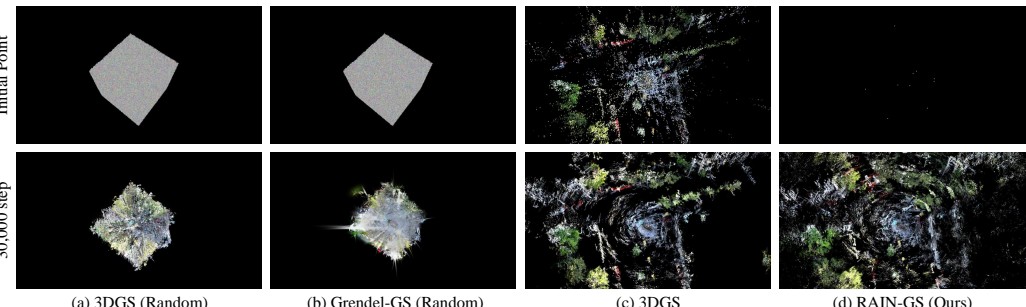

(a) 3DGS (Random)   (b) Grendel-GS (Random)   (c) 3DGS   (d) RAIN-GS (Ours)

Figure 14: **Displacements of Gaussians from initial positions including Grendel-GS (Zhao et al., 2024).**

## F.5 Analysis of Movement of Each Gaussian in RAIN-GS.

Through the extended analysis of Section A.2 and Table 2, we assess how effectively RAIN-GS mitigates the transportability issues associated with conventional Gaussian Splatting methods. By combining our methods (SLV initialization, progressive Gaussian low-pass filtering, and ABE-Split) RAIN-GS successfully mitigates the problem of lack of transportability. Starting from very sparse number of Gaussians, RAIN-GS can successfully transport the Gaussians to where the scene is located. For quantitative comparisons of the Gaussians movements shown in Table 2, we provide the overall movements of RAIN-GS in the Mip-NeRF 360 dataset in Table 13. As shown in the analysis, RAIN-GS shows the largest movements throughout the overall optimization process. This is straightforward as starting from random indicates that the initial point clouds need to move more to fully represent the scene.

## F.6 Additional analysis on Grendel-GS

In this section, we further investigate if other training strategies can mitigate the limitation of the lack of transportability of Gaussians. Specifically, we analyze Grendel-GS (Zhao et al., 2024), which introduces batch-wise training for the 3DGS optimization. We extend our previous analysis of evaluating the total movements of the Gaussians throughout the optimizations in the Mip-NeRF360 dataset. Table 14 reveals that batch-wise training slightly mitigates the limitation of the original 3DGS, showing larger average movements when compared to 3DGS (Random). However, even with larger movements it still shows smaller movements when compared to 3DGS (SfM) which indicates the lack of sufficient movement as mentioned in Section 4.2. Figure 14 shows that similar to 3DGS (Random), the learned Gaussians fail to learn the structure of the scene, maintaining the bounding-box like shape even after optimization. Figure 15 shows that due to the lack of sufficient movement, the Gaussians fail to model the house in the red bounding box that is located in a distant region. This additional analysis reveals the effectiveness of our approach, where both Table 13 and Figure 14 verify that our approach robustly learns the structure of the scene showing large movements.

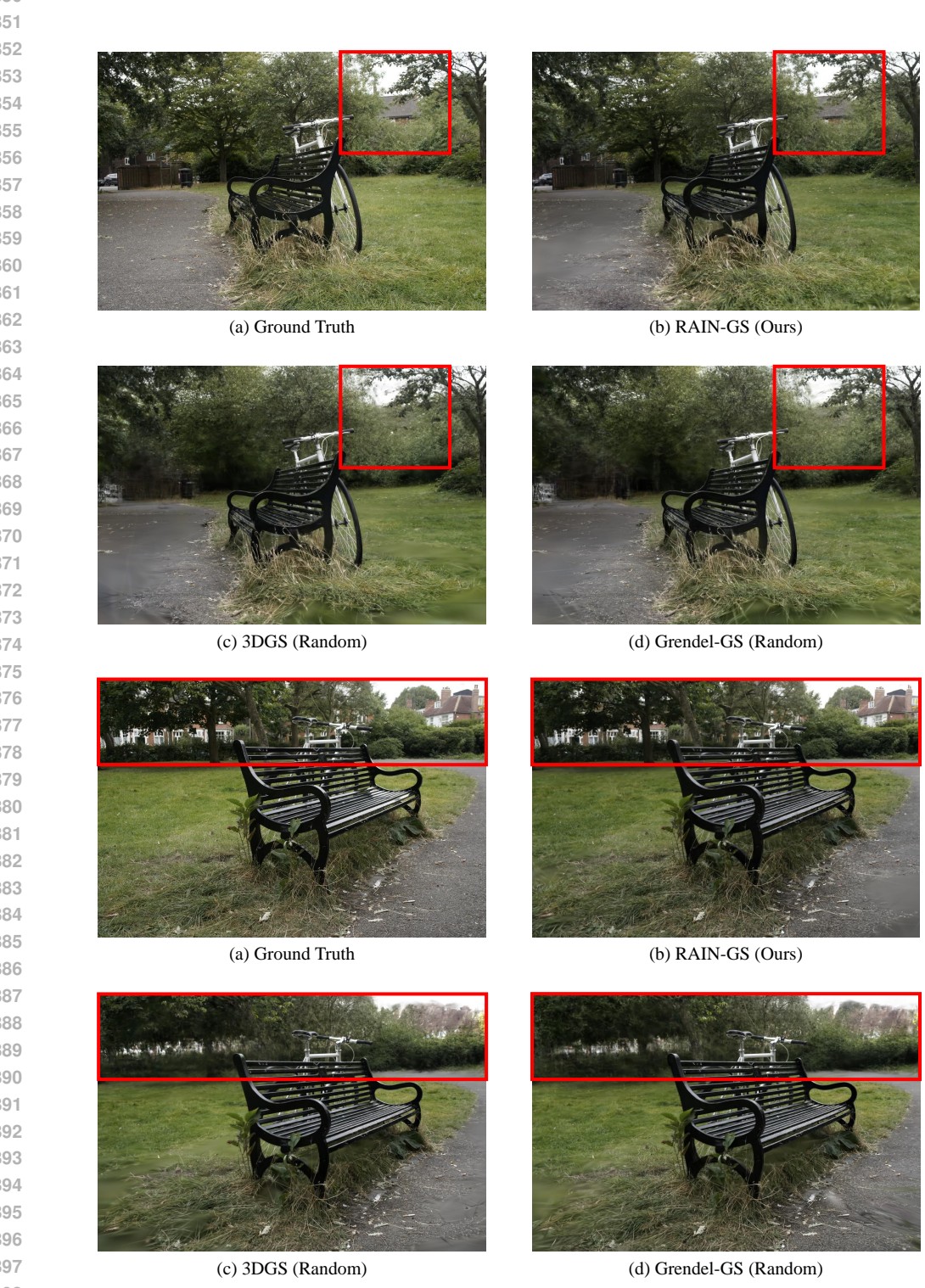

Figure 15: **Qualitative comparisons on 'bicycle' scene rendered using different initial point clouds including Grendel-GS (Zhao et al., 2024).** The red-bounding box region shows examples of under-reconstruction, where the house in the background remains un-reconstructed.

|        | 3DGS (SfM) | 3DGS (Noisy SfM) | 3DGS (Random) | Grendel-GS (Random) | RAIN-GS (Ours) |
|--------|-----------|------------------|---------------|---------------------|----------------|
| Means  | 0.704     | 0.650            | 0.395         | 0.641               | 16.403         |
| Stds   | 2.207     | 0.729            | 0.402         | 0.678               | 14.606         |
| Top 1% | 10.755    | 3.646            | 1.923         | 2.760               | 68.919         |

Table 14: **Movement of Gaussians including Grendel-GS (Zhao et al., 2024).**

