# OpenReview forum: "Relaxing Accurate Initialization Constraint for 3D Gaussian Splatting"
_ICLR.cc/2025/Conference — Submitted to ICLR 2025_

### Official Review · Reviewer_gqEX · 2024-10-21

**Soundness:** 2
**Presentation:** 2
**Contribution:** 3
**Rating:** 6
**Confidence:** 4

**Summary:**

This paper aims to alleviate the critical requirements for accurate initialization for 3DGS. The proposed RAIN-GS attributes the problem to limited Gaussian transportability. Based on this observation, it introduces a novel initialization strategy and adaptive density control strategy, as well as a progressive Gaussian low-pass filtering to resolve the problem.

**Strengths:**

1. Significance. Relaxing the initialization constraint of 3DGS is a critical and meaningful problem. The RAIN-GS realizes an impressive performance without the need of SfM points, proving it is an efficient solution.

**Weaknesses:**

1. As a submission to ICLR, it's not reasonable to only include 3DGS and Mip-Splatting for comparison. 2DGS sacrifices representation ability for more accurate geometry. Thus it's meaningless and unfair to compare with it purely on rendering quality. A more reasonable choice should be ScaffoldGS [CVPR24] or other improvements on ECCV24 or NIPS24, especially when they are randomly initialized.
2. The authors claim limited Gaussian transportation as a reason for the problem, and provide the analysis on movement of Gaussians as evidence. However, it remains unclear where do RAINGS's improvement comes from. The author should compare their method in both Table 2 and Figure 2 to validate the motivation and effectiveness of their method.

**Questions:**

1. The author claims that the limited Gaussian transportability is mainly due to suboptimal performance caused by exclusive reliance on image photometric loss. However, is there any possibility that the suboptimal results are from batch size 1 optimization? Can this problem be well resolved by simply using a parallel training paradigm like GrendelGS?
2. In Table 5, the author claims that when combined with the low-pass filter and ABE-Split algorithm, SLV initialization shows the best performance. However, there seems to be a lack of an experiment combining the proposed low-pass filter, N=100K, and ABE-Split to provide a comparison.

---

> ### Author Response · Authors · 2024-11-24
> **Response to Reviewer gqEX (1/2)**
>
> > As a submission to ICLR, it's not reasonable to only include 3DGS and Mip-Splatting for comparison. 2DGS sacrifices representation ability for more accurate geometry. Thus it's meaningless and unfair to compare with it purely on rendering quality. A more reasonable choice should be ScaffoldGS [CVPR24] or other improvements on ECCV24 or NIPS24, especially when they are randomly initialized.
> >
>
> We would like to first clarify that the methods we include for comparisons all assume that accurate point clouds are readily available. Therefore, we have included the performance of these methods trained from SfM point clouds as reference to show the robustness of our strategy when being trained from random point clouds. As suggested from the reviewer, we have conducted extra experiments with Scaffold-GS[1] on the Mip-NeRF360 dataset [2]. From our experiments, Scaffold-GS also shows large performance drops when trained from random point clouds.
>
> In addition, we found that the positions of the Gaussians achieved from training 3DGS with our strategy can replace the original SfM point clouds. From this observation, we have conducted additional experiments of replacing initial point clouds with ours for training Scaffold-GS, which was suggested by reviewer pLoa. As RAIN-GS can be trained from random point clouds, replacing the initial point cloud with ours enable Scaffold-GS to also be trained from random point clouds. The details of our experiments are as follows:
>
> 1. We train 3DGS with random point clouds using our strategy. For time and memory efficiency, we train for only 7000 steps. We also found that the number of Gaussians at 7000 steps are similar to the number of initial point clouds achieved with SfM in the Mip-NeRF360 dataset allowing fair comparison.
> 2. We save the positions of the Gaussians as point clouds.
> 3. From the saved point clouds, we train Scaffold-GS.
>
> The performance is shown below:
>
> | Methods | bicycle | flowers | garden  | stump | treehill | room | counter | kitchen | bonsai | Average |
> | --- | --- | --- | --- | --- | --- | --- | --- | --- | --- | --- |
> | Scaffold-GS (SfM) | 24.50 | 21.44 | 27.17 | 26.27 | **23.15** | **31.93** | 29.34 | 31.30 | **32.70** | 27.53 |
> | Scaffold-GS (Random) | 23.05 | 19.79 | 26.38 | 22.48 | 21.33 | 30.65 | 28.15 | 29.43 | 30.66 | 25.77 |
> | **Scaffold-GS + RAIN-GS (Ours)** | **25.32** | **21.82** | **27.75** | **26.45** | 23.06 | **31.93** | **29.57** | **31.82** | 32.21 | **27.77** |
>
> We observe that leveraging the point clouds achieved from RAIN-GS even outperforms the results of Scaffold-GS trained with SfM point clouds, successfully enabling Scaffold-GS to be trained from random point clouds. This reveals an additional downstream application of our strategy. Even with accurate camera poses and point clouds achieved from SfM algorithms, utilizing the point clouds from RAIN-GS can further boost the performance of various 3DGS extensions.
>
> > The authors claim limited Gaussian transportation as a reason for the problem, and provide the analysis on movement of Gaussians as evidence. However, it remains unclear where do RAINGS's improvement comes from. The author should compare their method in both Table 2 and Figure 2 to validate the motivation and effectiveness of their method.
> >
>
> We thank the reviewer for pointing this out. By combining our methods (SLV initialization, progressive Gaussian low-pass filtering, and ABE-Split) RAIN-GS successfully mitigates the problem of lack of transportability. We kindly direct the reviewer’s attention to Figure 7 in the Appendix where the qualitative improvement of RAIN-GS can be seen in the same ‘Truck’ scene. Starting from very sparse number of Gaussians, RAIN-GS can successfully transport the Gaussians to where the scene is located. For quantitative comparisons of the Gaussians movements shown in Table 2, we provide the overall movements of RAIN-GS in the Mip-NeRF 360 dataset as follows:
>
> |  | SfM | Noisy SfM | Random | **RAIN-GS (Ours)** |
> | --- | --- | --- | --- | --- |
> | Means | 0.704 | 0.650 | 0.395 | **16.403** |
> | Stds | 2.207 | 0.729 | 0.402 | **14.606** |
> | Top 1% | 10.755 | 3.646 | 1.923 | **68.919** |
>
> As shown in the analysis, RAIN-GS shows the largest movements throughout the overall optimization process. This is straightforward as starting from random indicates that the initial point clouds need to move more to fully represent the scene.
>
> ---
>
> [1] Lu, Tao, et al. "Scaffold-gs: Structured 3d gaussians for view-adaptive rendering." Proceedings of the IEEE/CVF Conference on Computer Vision and Pattern Recognition. 2024.
>
> [2] Barron, Jonathan T., et al. "Mip-nerf 360: Unbounded anti-aliased neural radiance fields." Proceedings of the IEEE/CVF conference on computer vision and pattern recognition. 2022.

---

> ### Author Response · Authors · 2024-11-24
> **Response to Reviewer gqEX (2/2)**
>
> > The author claims that the limited Gaussian transportability is mainly due to suboptimal performance caused by exclusive reliance on image photometric loss. However, is there any possibility that the suboptimal results are from batch size 1 optimization? Can this problem be well resolved by simply using a parallel training paradigm like GrendelGS?
> >
>
> We would like to clarify that although batch training can improve the overall training speed of 3DGS and robustness, the limitation of limited Gaussian transportability cannot be directly addressed. Limited Gaussian transportability mainly comes from Gaussians modeling very local regions, which does not change by training with increased batch sizes. Nevertheless, we provide the results of GrendelGS[1] trained with random point clouds in the Mip-NeRF360 dataset [2] with an effective batch size of 16 following its default arguments, which also show similar performance drops.
>
> | Methods | bicycle | flowers | garden  | stump | treehill | room | counter | kitchen | bonsai | Average |
> | --- | --- | --- | --- | --- | --- | --- | --- | --- | --- | --- |
> | 3DGS (Random) | 23.781 | 20.450 | 26.417 | 23.067 | 21.456 | 29.987 | 27.963 | 30.353 | 29.562 | 25.893 |
> | Grendel-GS (Random) | 23.918 | 20.523 | 26.357 | 22.866 | 21.443 | 29.615 | 27.130 | 29.986 | 29.625 | 25.718 |
> | **RAIN-GS (Ours)** | **25.373** | **22.118** | **27.277** | **27.029** | **22.887** | **30.866** | **28.681** | **31.416** | **31.610** | **27.473** |
>
> > In Table 5, the author claims that when combined with the low-pass filter and ABE-Split algorithm, SLV initialization shows the best performance. However, there seems to be a lack of an experiment combining the proposed low-pass filter, N=100K, and ABE-Split to provide a comparison.
> >
>
> We apologize for the lack of detailed ablations of our components. We provide a more detailed ablation of each of our components below.
>
> | Low-pass filter | Init. | ABE-Split | PSNR $\uparrow$ | SSIM $\uparrow$  | LPIPS $\downarrow$ |
> | --- | --- | --- | --- | --- | --- |
> | Constant | N=100$K$ | $\times$ | 25.893 | 0.764 | 0.273 |
> | Constant | N=100$K$ | $\checkmark$ | 26.970 | 0.805 | 0.227 |
> | Constant  | SLV | $\times$ | 25.815 | 0.759 | 0.280 |
> | Constant  | SLV | $\checkmark$ | 26.395 | 0.785 | 0.231 |
> | **Ours** | **N=100$K$** | $\times$ | **26.116** | **0.765** | **0.273** |
> | **Ours**  | **N=100$K$** | **$\checkmark$** | **26.982** | **0.808** | **0.226** |
> | Ours | SLV | $\times$ | 26.288 | 0.769 | 0.273 |
> | Ours | SLV | $\checkmark$ | 27.473 | 0.818 | 0.215 |
>
> ---
>
> [1] Zhao, Hexu, et al. "On scaling up 3d gaussian splatting training." *arXiv preprint arXiv:2406.18533* (2024).
>
> [2] Barron, Jonathan T., et al. "Mip-nerf 360: Unbounded anti-aliased neural radiance fields." Proceedings of the IEEE/CVF conference on computer vision and pattern recognition. 2022.

---

> ### Comment · Reviewer_gqEX · 2024-11-25
>
> I appreciate the authors' effort in offering extensive experiments. My concerns regarding the second weakness and the second question have been resolved. However, for the first weakness, the concern about the unfairness of comparing with 2DGS is not related to input and remains unresolved. To compare with ScaffoldGS, it's not reasonable to compare at 7K iterations when this experiment should be used for Table 3 of the main paper (I trust that over a week is sufficient enough to finish the 30K version experiments). It's also weird to base ScaffoldGS on a pre-trained point cloud to show adaptability. I trust even a vanilla 3DGS trained with 7k can improve the performance. An appropriate way should be to incorporate the technique into ScaffoldGS and finish full training.
>
> For the first question, the author's conclusion is not convincing. On the one hand, pure metrics comparison is not solid evidence that transportability can't be resolved by >1 batch size. A micro-analysis is necessary. On the other hand, to show whether parallel training helps, the training iteration should be aligned, and compare transportability among methods at the same iteration.
>
> Therefore, the concern is far from fully resolved. And the score will be kept.

---

> > ### Author Response · Authors · 2024-11-25
> > **Further clarifications of remaining concerns (2/2)**
> >
> > 3. **Analysis of Grendel-GS**
> >
> > We agree with the reviewer’s suggestion that a similar micro-analysis of Grendel-GS would strengthen our analysis and findings. We show the movement analysis of Grendel-GS when trained from random point clouds for 30K iterations below:
> >
> > |  | 3DGS (SfM) | 3DGS (Noisy SfM) | 3DGS (Random) | Grendel-GS (Random) | **RAIN-GS (Ours)** |
> > | --- | --- | --- | --- | --- | --- |
> > | Means | 0.704 | 0.650 | 0.395 | 0.641 | **16.403** |
> > | Stds | 2.207 | 0.729 | 0.402 | 0.678 | **14.606** |
> > | Top 1% | 10.755 | 3.646 | 1.923 | 2.760 | **68.919** |
> >
> > We found that aligned with the assumption of the reviewer, Grendel-GS shows larger movements when compared to the movements calculated in 3DGS (Random). We apologize for the confusion of providing a wrong initial analysis. From a more detailed analysis suggested by the reviewer, we noticed that the batch training strategy can also be utilized to mitigate the issue of lack of transportability. However, from a similar qualitative analysis we have shown in Figure 1 and Figure 7, we have found that batch training alone is not enough to fully address the limitation of lack of transportability. We have visualized these additional analysis in Figure 14 and Figure 15 in the Appendix F.6.
> >
> > - The comparison in Figure 14 shows that even with larger movements, Grendel-GS trained from random point clouds fail to learn the structure of the scene and maintain the structure of the bounding box the initial points were initialized.
> > - The comparison in Figure 15 also shows that even with larger movements, the movement is not sufficient enough to correctly model regions that are in distant areas, failing to model the house in the red bounding box.
> >
> > These results show that the batch training of Grendel-GS is not enough and verifies the effectiveness of our approach. We thank the reviewer for the valuable feedback which can further strengthen our work’s contributions.
> >
> > We are grateful for the opportunity to clarify and hope our additional explanations can resolve the remaining concerns of the reviewer.

---

> ### Author Response · Authors · 2024-11-25
> **Further clarifications of remaining concerns (1/2)**
>
> We thank the reviewer for the quick response and we are delighted that our response addressed some of the reviewer’s concerns. We would like to further clarify the remaining concerns as we have noticed that some of experiments have been misunderstood by the reviewer.
>
> 1. **2DGS as a comparison**
>
> We would like to clarify that as our method (RAIN-GS) builds above the vanilla-3DGS, the main comparison and strengths of our method should be mainly compared with the performance of vanilla-3DGS. Therefore, we did not claim the superiority over 2DGS or Mip-Splatting as the strength of our work. However, we acknowledge the suggestion of the reviewer and we will exclude the comparisons with 2DGS and replace it with ScaffoldGS.
>
> 2. **Additional experiments with ScaffoldGS**
> - **Training Iterations**
>
> We apologize for the confusion caused in our explanation. First of all, the performance of Scaffold-GS with SfM points, random points, and points from RAIN-GS is the performance after training for 30K steps. Only RAIN-GS is trained for 7K steps. This is also for the case of our experiments with Grendel-GS as we report the values after training for 30K steps.
>
> - **Initial point clouds**
>
> As the reviewer mentioned, more initial points for point clouds or learned parameters for the Gaussians can provide extra performance boosts. Therefore, we have chosen 7K steps as the number of Gaussians after 7K steps is similar to the number of initial points from SfM. In addition, we only save the positions of the Gaussians as point clouds leaving other parameters unsaved. Note that this is even less beneficial compared to utilizing SfM point clouds as they also save the colors of the points, which are then used to initialize the SH coefficients of the Gaussians at the beginning of training.
>
> > I trust even a vanilla 3DGS trained with 7k can improve the performance. An appropriate way should be to incorporate the technique into ScaffoldGS and finish full training.
>
> As mentioned above, as the experiments for Scaffold-GS + RAIN-GS is done by simply replacing the initial point clouds from SfM or random to RAIN-GS, this experiment reveals how well RAIN-GS has learned the structures of the scene after being trained from random point clouds. As we visualized and evaluated on Figure 2 and Table 2, vanilla-3dgs trained from random point clouds fail to learn the structure, due to the lack of transportability. From our analysis, it can be assumed that vanilla-3dgs (random) cannot provide any benefits. Nevertheless, we show the results of replacing the initial point clouds of Scaffold-GS with the point clouds saved from vanilla-3dgs (random) for 7K steps.
>
> The results are shown below:
>
> Note that Scaffold-GS (SfM), Scaffold-GS (Random), Scaffold-GS + 3DGS (Random), and Scaffold-GS + RAIN-GS (Ours) is all trained for 30K iterations and only the step of achieving point clouds from RAIN-GS and 3DGS from random point clouds is done for 7K iterations.
>
> | Methods | bicycle | flowers | garden  | stump | treehill | room | counter | kitchen | bonsai | Average |
> | --- | --- | --- | --- | --- | --- | --- | --- | --- | --- | --- |
> | Scaffold-GS (SfM) | 24.50 | 21.44 | 27.17 | 26.27 | **23.15** | **31.93** | 29.34 | 31.30 | **32.70** | 27.53 |
> | Scaffold-GS (Random) | 23.05 | 19.79 | 26.38 | 22.48 | 21.33 | 30.65 | 28.15 | 29.43 | 30.66 | 25.77 |
> | Scaffold-GS + 3DGS (Random) | 23.24 | 19.94 | 26,39 | 22.53 | 20.92 | 31.37 | 28.72 | 30.17 | 28.39 | 25.66 |
> | **Scaffold-GS + RAIN-GS (Ours)** | **25.32** | **21.82** | **27.75** | **26.45** | 23.06 | **31.93** | **29.57** | **31.82** | 32.21 | **27.77** |
>
> Our experiments reveal that point clouds achieved from vanilla-3dgs does not improve Scaffold-GS. The experiment also reveals two applications of our method for other 3DGS methods:
>
> - **When accurate initial point clouds are not available :** As RAIN-GS starts from random point clouds, if other methods utilize the point clouds saved from 7K iterations of RAIN-GS, this enables other methods to be robustly trained even in settings where SfM point clouds are not available.
> - **When accurate initial point clouds are available :** As the performance of Scaffold-GS + RAIN-GS is stronger than Scaffold-GS + SfM, other methods can utilize our method for better performance even when SfM point clouds are available.
>
> > An appropriate way should be to incorporate the technique into ScaffoldGS
> >
>
> Although our method can be applied to various 3DGS extensions that leverage the original optimization algorithm of vanilla-3dgs in a plug-and-play manner, we would like to clarify that it is not possible for the case of Scaffold-GS as they modify the optimization algorithm itself. Instead of optimizing the Gaussians directly, they optimize the voxel grid, which is constructed from the initial point clouds. Therefore, we have shown an additional direction of how our method can be beneficial for 3DGS extensions (Scaffold-GS) that modify the optimization algorithm in our rebuttal.

---

> ### Comment · Reviewer_gqEX · 2024-11-27
>
> I appreciate the authors' effort in providing additional experiments. The experiments related to GrendelGS have resolved my concern and show an interesting conclusion. This part validates the effectiveness and necessity of RAIN-GS from a meaningful viewpoint and the conclusion deserves to be mentioned in the main body of the paper. The explanation regarding Scaffold GS is to some extent chaotic, especially about the experiment setting. For instance, "Scaffold-GS + RAIN-GS (Random)" seems to be a better replacement for "Scaffold-GS + RAIN-GS (Ours)". The problem of implementing the same technique to Scaffold-GS should also be emphasized at very first. But I acknowledge that approximately, the effectiveness has also been validated with these extra experiments.
>
> In a nutshell, the remaining concerns have been resolved. I would like to raise my score to slightly above borderline with some parts of the paper being revised properly.

---

> > ### Author Response · Authors · 2024-11-28
> >
> > We are delighted to find that our responses have addressed the reviewer's concerns. We apologize for any of our initial explanations that caused additional confusion, and we will focus on clarifying our experimental settings, contribution, and inputs as the reviewer suggested. We thank the reviewer for all the valuable feedback and positive evaluation of our work.

---

### Official Review · Reviewer_Caqa · 2024-11-01

**Soundness:** 3
**Presentation:** 3
**Contribution:** 3
**Rating:** 6
**Confidence:** 4

**Summary:**

This paper aims to reconstruct Gaussian scenes using randomly initialized point clouds. The designs guide the Gaussian scenes to first establish a global structure, thereby avoiding local optima. Although relatively simple, this method even achieves superior reconstruction results in certain scenes compared to Colmap+3DGS. However, more experiments and analysis are needed to support the motivation.

**Strengths:**

1. This paper proposes to reconstruct Gaussian scenes using randomly initialized point clouds. In some cases, the proposed method surpasses Colmap+3DGS, which is impressive.
2. The proposed method encourages Gaussian scenes to first learn global structural information before focusing on detailed information.
3. The analysis in this paper is well-organized and clearly written.

**Weaknesses:**

1. The design in this paper mainly comes from experimental results. While the approach is simple and effective, it lacks theoretical novalities, and many hyperparameters require tuning. It would be beneficial for the authors to provide more detailed ablation experiments on these hyperparameters.

2. The paper assumes known multi-view image poses without point cloud reconstruction, but further analysis and explanation are needed regarding in which case this setting makes scene:
- If the image poses are obtained from sensors, how does the method handle pose errors? How much error can it handle? Should the poses being optimized to improve robustness?
- If the multi-view images are generated from generative models, such as text-to-3D models, please validate the effectiveness of the proposed method in these cases.

**Questions:**

1. In Line 431, the authors mention that the reconstruction results are based on poses from sensors. Does the the pose errors impact the final reconstruction performance, and can the proposed method mitigate this issue?

2. As shown in Figures 1 and 3, why does the improvement mainly occur in edge regions or distant areas?

3. Some further discussions: With the recent advancements in monocular reconstruction algorithms, such as Dust3R and DepthPro, which allow for effective 3D reconstruction without relying on Colmap or even poses, would the proposed method conflict with or complement these approaches? Could case studies be conducted to explore this?

I might raise my rating if my concerns are addressed.

---

> ### Author Response · Authors · 2024-11-24
> **Response to Reviewer Caqa (1/4)**
>
> We thank the reviewer for the thorough review and valuable feedback that can improve our work when addressed. Our detailed response can be seen below:
>
> > The design in this paper mainly comes from experimental results. While the approach is simple and effective, it lacks theoretical novalities, and many hyperparameters require tuning. It would be beneficial for the authors to provide more detailed ablation experiments on these hyperparameters.
> >
>
> We would like to first clarify that the vanilla 3DGS has significant amount of hyperparameters that are carefully tuned and among these we have only modified the hyperparameters that are effected by our proposed strategy while keeping others unchanged. The changed or newly introduced hyperparameters are:
>
> - Initial number of Gaussians for SLV initialization
> - Schedule for Progressive Gaussian Low-Pass filtering
> - Division factor of Gaussians during the splitting process
>
> We show the ablations of each hyperparameters below:
>
> 1. Starting with N=10 for SLV initialization.
>
> Our motivations are to initialize sparse Gaussians with large variance which enables the Gaussians to receive sufficient amount of gradients during training. We show the ablation of starting with different number of Gaussians when training our method below. We show the average performance in Mip-NeRF360 dataset.
>
> | $N$ | 100$K$ | 1$K$ | 100 | 10 (Ours) |
> | --- | --- | --- | --- | --- |
> | PSNR $\uparrow$ | 26.982 | 27.157 | 27.257 | 27.473 |
>
> We observe that as we decrease the number of $N$, which lead to initializing more sparse and large variance Gaussians, the performance improves. We select $N=10$ as default in our framework due to best performance. Note that, as shown in the original ablation of initial number of Gaussians in Table 5, using SLV initialization only shows performance improvements when utilized with our other components.
>
> 1. The schedule for the progressive Gaussian low-pass filter control.
>
> We have analyzed different schedules where the ablations are shown in Appendix A.3.
>
> 1. Modifying the divide factor from 1.6 to 1.4
>
> Our motivations are to prevent the Gaussians from becoming too small, which can lead to similar problems of Gaussians only modeling very local regions. Therefore, we have reduces the divide factor to 1.4 to enable the Gaussians to have slightly larger sizes than 1.6. We show the ablation of different division factors below. We show the average performance in Mip-NeRF360 dataset.
>
> | Division factor | PSNR $\uparrow$ | SSIM $\uparrow$  | LPIPS $\downarrow$ |
> | --- | --- | --- | --- |
> | 1.5 | 27.168 | 0.805 | 0.229 |
> | 1.4 | 27.473 | 0.818 | 0.215 |
> | 1.3 | 27.109 | 0.805 | 0.233 |
>
> We select 1.4 as the default division factor due to its best performance. Note that we use all three hyperparameters as default for all datasets and evaluations.

---

> ### Author Response · Authors · 2024-11-24
> **Response to Reviewer Caqa (2/4)**
>
> > The paper assumes known multi-view image poses without point cloud reconstruction, but further analysis and explanation are needed regarding in which case this setting makes scene
> >
>
> Our strategy is effective and essential in the following settings:
>
> - Training with generated images using text-to-3D models
> - Noisy camera poses
> - Initial point clouds are noisy
>
> The detailed explanation of how our method can be applied is explained below:
>
> 1. **Reconstruction with generated images with text-to-3D generation models :** As suggested by the reviewer, we have conducted experiments of training 3DGS with multi-view images generated with DimensionX[1]. The generated images are readily available with camera pose but without any point clouds. Here we show that  Ours show significant improvements on PSNR over random point clouds. We also add qualitative results in Figure 5 of the revised pdf. It is worth noting that SfM algorithms [2] fail to converge in these scenes, making random point clouds the only choice for reconstruction.
>
>
>     | Methods / Scene | Scene1 | Scene2 |
>     | --- | --- | --- |
>     | Random | 11.17 | 12.27 |
>     | RAIN-GS (Ours) | **25.26** | **32.75** |
> 2. **Camera poses are noisy:** When the camera poses are achieved fully from sensors or SfM algorithms fail to provide accurate camera poses, it has been shown that the performance of both NeRF and 3DGS largely degrades [3,4]. To understand the robustness of our approach when the camera poses are noisy, we conducted extra experiments of training 3DGS with noisy camera poses from Mip-NeRF360 dataset [5], where the noisy camera poses are obtained by following the protocol of CamP [6], which adds approximately 5 degrees of noise to all the camera poses. Furthermore, as the original rasterizer of 3DGS does not propagate gradients to the camera poses, we have implemented a custom rasterizer for both 3DGS and ours to enable correction of the camera poses. We show both average results on Mip-NeRF360 dataset with and without camera pose correction below:
>
>
>     | Methods (w/o camera optimization) | PSNR $\uparrow$ | SSIM $\uparrow$  | LPIPS $\downarrow$ |
>     | --- | --- | --- | --- |
>     | Random | 16.026 | 0.386 | 0.623 |
>     | Ours | **16.492** | **0.404** | **0.607** |
>
>     | Methods (w/ camera optimization) | PSNR $\uparrow$ | SSIM $\uparrow$  | LPIPS $\downarrow$ |
>     | --- | --- | --- | --- |
>     | Random | 18.916 | 0.453 | 0.572 |
>     | Ours | **21.060** | **0.538** | **0.475** |
>
>     We observe that although noise in the camera pose largely degrades the performance in the Mip-NeRF360 dataset, Ours consistently achieve better results than the original random point cloud settings. The superior performance of Our method comes from the progressive Gaussian low-pass filtering and SLV initialization, where this approach is similar to the coarse-to-fine training adopted in settings of training noisy poses together with NeRF [3]. Combined with our experiments in Table1, Our approach can robustly train 3DGS in settings regardless of either the initial point cloud is noisy or the camera pose is noisy.
>
>
> ---
>
> [1] Sun, Wenqiang, et al. "DimensionX: Create Any 3D and 4D Scenes from a Single Image with Controllable Video Diffusion." arXiv preprint arXiv:2411.04928 (2024).
>
> [2] GLOMAP: Global Structure-from-Motion Revisited
>
> [3] Lin, Chen-Hsuan, et al. "Barf: Bundle-adjusting neural radiance fields." Proceedings of the IEEE/CVF international conference on computer vision. 2021.
>
> [4] Fu, Yang et al. “COLMAP-Free 3D Gaussian Splatting.” *2024 IEEE/CVF Conference on Computer Vision and Pattern Recognition (CVPR)* (2023): 20796-20805.
>
> [5] Barron, Jonathan T., et al. "Mip-nerf 360: Unbounded anti-aliased neural radiance fields." Proceedings of the IEEE/CVF conference on computer vision and pattern recognition. 2022.
>
> [6] Park, Keunhong, et al. "Camp: Camera preconditioning for neural radiance fields." ACM Transactions on Graphics (TOG) 42.6 (2023): 1-11.

---

> ### Author Response · Authors · 2024-11-24
> **Response to Reviewer Caqa (3/4)**
>
> Continued from the previous response, an additional setting where our strategy can be used is when the initial point clouds include noise.
>
> **Initial point clouds include noise:** As shown in Table 1, adding a small noise $\epsilon$ sampled from a normal distribution of $N(0,0.5)$ directly degrades the performance of 3DGS (-0.4dB), due to its large dependency of accurate initial point clouds. In real-world applications, as the user cannot evaluate how accurate the point clouds achieved from SfM algorithms are, Our method can be applied to avoid unwanted performance degradation.
>
> In addition, we found that the positions of the Gaussians achieved from training 3DGS with our strategy can replace the original SfM point clouds. We have conducted additional experiments of replacing SfM point clouds with ours for training Scaffold-GS [1], which was suggested by reviewer pLoa. Specifically, we replace the initial point clouds with ours by the following steps:
>
> 1. We train 3DGS with random point clouds using our strategy. For time and memory efficiency, we train for only 7000 steps. We also found that the number of Gaussians at 7000 steps are similar to the number of initial point clouds achieved with SfM in the Mip-NeRF360 dataset[2] allowing fair comparison.
> 2. We save the positions of the Gaussians as point clouds.
> 3. From the saved point clouds, we train Scaffold-GS for 30K iterations.
>
> The performance is shown below:
>
> Note that Scaffold-GS (SfM), Scaffold-GS (Random), and Scaffold-GS + RAIN-GS (Ours) is all trained for 30K iterations and only the step of achieving point clouds from RAIN-GS is done for 7K iterations.
>
> | Methods | bicycle | flowers | garden  | stump | treehill | room | counter | kitchen | bonsai | Average |
> | --- | --- | --- | --- | --- | --- | --- | --- | --- | --- | --- |
> | Scaffold-GS (SfM) | 24.50 | 21.44 | 27.17 | 26.27 | **23.15** | **31.93** | 29.34 | 31.30 | **32.70** | 27.53 |
> | Scaffold-GS (Random) | 23.05 | 19.79 | 26.38 | 22.48 | 21.33 | 30.65 | 28.15 | 29.43 | 30.66 | 25.77 |
> | **Scaffold-GS + RAIN-GS (Ours)** | **25.32** | **21.82** | **27.75** | **26.45** | 23.06 | **31.93** | **29.57** | **31.82** | 32.21 | **27.77** |
>
> We observe that even with camera poses and point clouds achieved from SfM algorithms, utilizing the point clouds from RAIN-GS (Ours) can further improve the performance of various 3DGS extensions.
>
> All three settings explain the effectiveness of our approach in real-world scenarios.
>
> ---
>
> [1] Lu, Tao, et al. "Scaffold-gs: Structured 3d gaussians for view-adaptive rendering." Proceedings of the IEEE/CVF Conference on Computer Vision and Pattern Recognition. 2024.
>
> [2] Barron, Jonathan T., et al. "Mip-nerf 360: Unbounded anti-aliased neural radiance fields." Proceedings of the IEEE/CVF conference on computer vision and pattern recognition. 2022.

---

> ### Author Response · Authors · 2024-11-24
> **Response to Reviewer Caqa (4/4)**
>
> > As shown in Figures 1 and 3, why does the improvement mainly occur in edge regions or distant areas?
> >
>
> We thank the reviewer for pointing this out. Your observation is correct that our approach can largely improve regions that are located in distant areas. It is because the main contribution of our method is to mitigate the lack of transportability of Gaussians, which prevent the Gaussians from successfully modeling areas that are far away. This becomes beneficial in overall reconstruction quality, mainly due to enabling each Gaussians to only properly model regions that are close in their 3D locations. Specifically, consider a Gaussian that is modeling a edge of a tree and a sky in the pixel space. If there is no Gaussian that can model the sky in 3D space, the Gaussian that is modeling the edge of a tree has to receive the supervision from both the tree and sky part of the pixel. We observe that by properly locating the Gaussians to model distant regions, each Gaussian can model the sky and tree independently, resulting in overall performance improvement including edge regions.
>
>
>
> > Some further discussions: With the recent advancements in monocular reconstruction algorithms, such as Dust3R and DepthPro, which allow for effective 3D reconstruction without relying on Colmap or even poses, would the proposed method conflict with or complement these approaches? Could case studies be conducted to explore this?
> >
>
> As mentioned by the reviewer, recent approaches have shown promising performance in estimating 3D geometry in a single feed-forward pass. We believe that our method can be used in a ***complementary manner*** with these advancements, mainly due to the robustness in both camera poses and point clouds compared to the original 3DGS trained with random point clouds.
>
> For the case of Dust3r [1], it has been shown that both pixel-level point maps and camera poses are estimated in a feed-forward manner. However, there are currently two problems that prevent Dust3r from replacing the traditional SfM algorithms.
>
> 1. **Time and Memory Complexity:** As the number of input images increase, Dust3r requires significant more time and memory due to its global alignment step. This prevents Dust3r from computing more than 100 images even in a H100 GPU.
> 2. **Pixel-level Pointmaps:** When initializing the Gaussians with point clouds, the estimated point maps from Dust3r is pixel-level which causes too many Gaussians to begin the optimization. In addition, it has been shown that the pointmaps from corresponding pixels are not perfectly aligned in 3D space, which can cause additional artifacts. Therefore, an additional method to sample a subset of these pointmaps are required to successfully train 3DGS. We believe that RAIN-GS can be an alternative, as RAIN-GS allows the training of 3DGS only with accurate camera poses. Simply by utilizing the camera poses from Dust3r will enable accurate reconstruction of 3DGS.
>
> For the case of DepthPro [2], as being a monocular depth estimator, it still struggles with estimating different scales for different images in the same scene, which can cause additional challenges of lifting points to 3D. In addition, DepthPro also estimates pixel-level depth maps which can cause similar problems with Dust3r, requiring an additional method to sample the points. In this case, RAIN-GS can also be used to allow efficient training without additional methods.
>
> ---
>
> [1] Wang, Shuzhe, et al. "Dust3r: Geometric 3d vision made easy." *Proceedings of the IEEE/CVF Conference on Computer Vision and Pattern Recognition*. 2024.
>
> [2] Bochkovskii, Aleksei, et al. "Depth pro: Sharp monocular metric depth in less than a second." *arXiv preprint arXiv:2410.02073* (2024).

---

> ### Author Response · Authors · 2024-11-26
>
> Dear Reviewer Caqa,
>
> As the author-reviewer discussion period is coming to an end, we wanted to kindly remind you of our responses to your comments. We greatly value your feedback and are eager to address any additional questions or concerns you might have.
>
> Please let us know if there's any further information we can provide to facilitate the discussion process.
>
> We are highly appreciated for your time and consideration.

---

> ### Comment · Reviewer_Caqa · 2024-11-26
> **Further response to the author's reply**
>
> Thank you for the detailed response! I still have some concerns:
>
> 1. Main concern: The author provided results with "Camera poses are noisy." While the proposed method slightly outperforms random initialization, its performance drops significantly under noisy poses. Considering that noisy sensor poses is a common case within none-SFM assumption, the reviewers encourage the author to integrate pose refinement into the framework to make the method more comprehensive.
>
> 2. Other concern: The author provided results combined with DimensionX. Although the author mentioned that SFM might fail in these scenarios. However, MVS can be utilized to provide a more accurate initialization with known image poses.
>
> Based on the considerations above, I will keep my rating.

---

> > ### Comment · Reviewer_pLoa · 2024-11-27
> > **About noisy camera poses**
> >
> > I'm also interested in applying the method to scenes with noisy camera poses, but I think it should be another paper that try to solve training 3DGS with noisy camera poses. The work is still useful when we have accurate camera poses but do not have sufficient point clouds (e.g. in texture-less areas, keypoint-based SfM/SLAM methods can not provide enough points).

---

> ### Author Response · Authors · 2024-11-26
> **Clarifying the contributions and focus of our work**
>
> We thank the reviewer for the response! We would like to further clarify the reviewer’s concerns. First of all, our work’s main contribution lies in investigating the core problem **“Why is 3DGS so dependent on the quality of initial point clouds even when trained from accurate camera poses?”.** From our analysis, we reveal that this behavior of 3DGS comes from the “lack of transportability” and also show that without changing the overall architecture of 3DGS, our proposed method RAIN-GS can effectively mitigate this limitation. This enables 3DGS to be trained from only accurate camera poses without being dependent on the quality of initial point clouds.
>
> > Performance drop when camera poses are noisy
> >
>
> It is worth noting that from our additional experiments shown in the rebuttal, training from noisy camera poses is **only one of the applications** of our method. The applications of our method include:
>
> - Training from text-to-3D generated cameras where SfM algorithms fail.
> - Training from noisy camera poses due to SfM algorithm failure.
> - An initial point cloud is noisy.
> - Providing better initial point clouds than SfM point clouds.
>
> In addition, we believe that the performance drop when training from noisy camera poses does not harm any of the contributions of our work. Compared to random point clouds, RAIN-GS shows much more improved results. The performance drop comes from training with noisy camera poses, which requires extra information or regularization (e.g., GT depth, Correspondence, and Monocular depth estimation models) to be more effectively mitigated as shown in [1,2]. As being one of our downstream applications, we believe that showing improvements over the baseline with no additional information or regularization verifies the effectiveness of our approach.
>
> > MVS can be utilized to provide a more accurate initialization with known image poses.
> >
>
> We would like to re-emphasize that the main contribution of our work is removing the dependency of 3DGS on accurate initial point clouds when accurate camera poses are available. When regarding NeRF [3] and its extensions, NeRF only requires accurate camera poses for high-quality novel view synthesis. We focus on removing this dependency as SfM algorithms takes a long processing time, often similar to the training time of 3DGS itself, which is impractical and inefficient for users already equipped with accurate camera poses. Although the suggested stage-wise training the reviewer has mentioned can be a possible approach, our work’s main contribution lies in **removing this one additional step** to successfully train 3DGS.
>
> We also want to kindly request the reviewer to add more information about the specific MVS technique the reviewer is referring to. To the best of our knowledge, there is no simple MVS technique that can be used to generate high-quality initial point clouds the vanilla-3DGS can be successfully trained. In our experiments, SfM algorithms have failed to converge, failing to provide any accurate camera poses. Other choices can be utilizing NeRF variants as NeRF only requires accurate camera poses, which takes long training times and the complex task of transforming trained densities of NeRFs to point clouds. As a result, we do not believe there is an efficient and straightforward approach to generating initial point clouds for the vanilla-3DGS, further verifying the contribution of removing the dependency of 3DGS on accurate initial point clouds.
>
> ---
>
> [1] Fu, Yang et al. “COLMAP-Free 3D Gaussian Splatting.” *2024 IEEE/CVF Conference on Computer Vision and Pattern Recognition (CVPR)* (2023): 20796-20805.
>
> [2] Jiang, Kaiwen, et al. "A construct-optimize approach to sparse view synthesis without camera pose." *ACM SIGGRAPH 2024 Conference Papers*. 2024.
>
> [3] Mildenhall, Ben, et al. "Nerf: Representing scenes as neural radiance fields for view synthesis." Communications of the ACM 65.1 (2021): 99-106

---

> ### Comment · Reviewer_Caqa · 2024-11-27
> **Response to the further replys**
>
> Thank you for the author's further response, but I am still not completely convinced. This mainly stems from considerations about the applicability of the paper. The motivation of the paper is the dependence of GS on accurate point clouds, so it is important to clarify the following points:
>
> 1. **Does SFM-generated point clouds introduce significant noise that affects GS?** The author demonstrated this through Scaffold-GS, but in Table 3 of the main text, the comparison between 3DGS (with SFM) and RAIN-GS doesn't show a very noticeable improvement.
>
> 2. **How to obtain accurate poses when SFM fails?**
>  - Poses come from sensors, but in this case, we need to consider sensor noise. The author's experiments confirm that this is necessary. Including this aspect would make the method more complete and practical.
>  - In 3D generation scenarios, accurate poses are known, and multi-view generation is performed, which seems to be the most reasonable assumption, but please refer to 3.
>  - Apart from these, could the author provide additional examples of scenarios where accurate poses can be obtained without using SFM?
> 3. **Do we have to use random point clouds when poses are known?** In text-to-3D generation cases where accurate poses are known, could MVS methods (such as MVSFormer++) generate point clouds more effectively to aid reconstruction (e.g., MVSGaussian)? A comparison of these methods in terms of both effectiveness and efficiency could help clarify this point.
>
> As a kind suggestion, I encourage the author to further clarify the paper's logic and reorganize the paper. Currently, the scenario of "none-SFM, accurate pose, only comparing with random point clouds" raises the above issues that should be addressed.

---

> > ### Author Response · Authors · 2024-11-27
> >
> > We thank the reviewer for the response and feedbacks. Yes, we agree with the reviewer that we should add some more detailed scenarios and explanations of “none-SfM, accurate pose” and how our method can be effectively utilized. During the rebuttal phase, we have shown multiple applications and we have added these explanations in the Appendix. We will further add more explanations to reduce any possible confusion.
> >
> > First of all, the most effective and essential application we have identified is when training images that are generated with text-to-3D images. As SfM algorithms failed to converge, without any additional step, 3DGS can only be trained with random point clouds. We would like to further explain the effectiveness over utilizing suggested MVS methods (MVSFormer++, and MVSGaussian).
> >
> > - **Generalized Networks for Reconstruction**
> >
> > As mentioned by the reviewer, there is an active line of research on training a generalized network that can be utilized for reconstruction such as Dust3r, and MVSFormer++. For Dust3r, as mentioned in our initial response, it requires very large computation power and additional sampling methods. For MVSFormer++, these methods are variants of generalized NeRF networks which aim to learn the prior of the scenes without the need for time-consuming per-scene optimization. However, the main performance bottleneck of these approaches is that they are still yet trained and tested in a single domain (indoor → indoor, outdoor → outdoor). Therefore, we found that these methods struggle in domain gaps such as handling generated synthetic images, where simply applying RAIN-GS with no additional step outperforms the initial point clouds achieved from MVSformer++. The results are shown below:
> >
> > | Methods / Scene | Scene1 | Scene2 |
> > | --- | --- | --- |
> > | Random | 11.17 | 12.27 |
> > | MVSFormer++ | 18.93 | 17.55 |
> > | RAIN-GS (Ours) | **25.26** | **32.75** |
> >
> > The results of 3DGS trained with point clouds from MVSFormer++ shows better results than training from purely random point clouds but still RAIN-GS outperforms in a large margin. The fundamental problem of these unwanted performance drops comes from the fact that **the user cannot evaluate how accurate the initial point clouds are**. All of our experiments highlight the need for an approach that can mitigate the dependency of 3DGS on the accuracy of initial point clouds. Although the improvements in the direction such as Dust3r, MVSFormer++ will reduce the problem of domain gaps, this also leads to 3DGS depending on the performance of these additional methods, which is **not addressing the fundamental problem itself.**
> >
> > - **Does SFM-generated point clouds introduce significant noise that affects GS?**
> >
> > For datasets that are captured in strict conditions such as Mip-NeRF360 datasets, we found that SfM point clouds are close to perfect. However, in real-world scenarios, we find that SfM cannot always be perfect. In our experiments, the results in Re10K show that 3DGS (SfM) shows lower performance than RAIN-GS similar to Table1 where introducing slight noise to the SfM results in lower performance (3DGS (SfM) vs 3DGS (Noisy SfM)). We believe that all of these problems stems from the large dependency of 3DGS on the quality of initial point clouds, although it is impossible to evaluate how accurate a initial point cloud is.
> >
> > - **How to obtain accurate poses when SFM fails?**
> >
> > In our rebuttal, we have summarized the settings where accurate poses are available when SfM fails.
> >
> > 1. **Reconstruction with generated images with text-to-3D generation models:** The experiments on RAIN-GS with generated images verify the effectiveness of our approach.
> > 2. **Reconstruction with camera poses achieved from graphic rendering programs such as Blender:** Images generated from Blender is readily available with accurate camera poses and due to the fact the original 3DGS shows the result of NeRF Synthetic dataset which is generated with Blender, from random point clouds. As random point clouds in [1] does not cause performance degradation due to the small scene that is bounded by the cameras, when dealing with larger scenes (e.g., Mip-NeRF360), applying Ours improve large performance over the random initialization setting. It has been shown that SfM algorithms fail in finding accurate correspondences in synthetic images, failing to converge [2].
> > 3. **Reconstruction with camera poses achieved from sensors (Re10K):** As shown in Table 4, Ours achieve better performance than both SfM and random point clouds. We also empirically find that with large texture-less regions, SfM algorithms can also fail to converge in these scenes.
> >
> > ---
> >
> > [1] Mildenhall, Ben, et al. "Nerf: Representing scenes as neural radiance fields for view synthesis." Communications of the ACM 65.1 (2021): 99-106
> >
> > [2] Chen, Yu, and Gim Hee Lee. "DReg-NeRF: Deep registration for neural radiance fields." Proceedings of the IEEE/CVF International Conference on Computer Vision. 2023.

---

> ### Comment · Reviewer_Caqa · 2024-11-27
> **Further response by the reviewer.**
>
> I appreciate the author's further response and additional results. I encourage the author to elaborate on these issues in the paper. At this point, I am willing to raise my score. My additional suggestion for this work is to incorporate pose refinement into the framework as an optional feature.

---

> > ### Author Response · Authors · 2024-11-27
> >
> > We thank the reviewer for all the valuable feedbacks and suggestions that can improve both the understanding and contributions of our work. As suggested by the reviewer, we will add a more detailed explanation and discussions on the paper. We are delighted to find that we have addressed the reviewer's concerns and thank the reviewer for the positive evaluation of our work.

---

### Official Review · Reviewer_pLoa · 2024-11-02

**Soundness:** 3
**Presentation:** 3
**Contribution:** 3
**Rating:** 6
**Confidence:** 5

**Summary:**

This paper aims to alleviate the initialization/densification problem in the original 3DGS, where regions are under-reconstructed in SfM. To solve the issue, the paper proposes initializing 3D Gaussians with larger variances so that Gaussians can cover more regions of the image and learn more about the global structure of the scene. Moreover, the paper proposes progressive low-pass filtering techniques to encourage Gaussians to receive more gradients during training. The paper also uses an adaptive bound-expanding split algorithm to overcome the limitation that the variance of Gaussians becomes small after multiple splitting steps. The experiments on the MipNeRF360 dataset and tanks-and-temples dataset show the effectiveness of the proposed method.

**Strengths:**

(1) The proposed method is simple to add to the existing 3DGS method

(2) The authors provide exhaustive experiments to verify the effectiveness of the proposed method

(3) The proposed method improves the original 3DGS clearly on the tanks-and-temples and the MipNeRF360 dataset.

**Weaknesses:**

(1) The ideas are intuitive and the contributions are not significant.

(2) While I appreciate their improvements to the original 3DGS methods, there are more variants of 3DGS, and I wonder would the same strategies proposed in this paper can improve the other 3DGS methods(such as scaffold-gs, etc).

(3) The method can improve 3DGS on the tanks-and-temples and MipNeRF360 dataset, I would like to see the effectiveness of this method on texture-less areas. And I would like to raise my score if the authors do so.

**Questions:**

The authors proposed the progressive low-pass filter to ensure 3D Gaussians receive more gradients during training. I wonder if is this equivalent to training 3D Gaussians from low-resolution images to high-resolution images in a coarse-to-fine manner. I would appreciate the authors to provide a fair comparison between the progressive low-pass filter and coarse-to-fine training strategy/

---

> ### Author Response · Authors · 2024-11-24
> **Response to Reviewer pLoa (1/2)**
>
> We thank the reviewer for the thorough review and valuable feedback that can improve our work when addressed. Our detailed response can be seen below:
>
> > The ideas are intuitive and the contributions are not significant.
> >
>
> We would like to clarify and re-emphasize our contributions as follows: Although both NeRF and 3DGS requires accurate camera poses for high-quality novel view synthesis, 3DGS has an additional requirement which is accurate initial point clouds. This unique requirement of 3DGS motivates the core contributions of our work analyzing **"Why and how does the accuracy of the initial point cloud affect the performance of 3DGS?”** through a comprehensive and in-depth analysis of the optimization process of 3DGS. From this analysis, we propose an efficient and effective method that successfully relaxes the constraint of accurate point clouds for 3D Gaussian Splatting.
>
> Our key contributions include:
>
> - **In-depth analysis of the dependency of 3DGS on accurate initial point clouds:** We analyze how the tendency of Gaussians falling into local minima affects the overall performance of 3DGS both qualitatively and quantitatively. We verify this behavior by analyzing the sensitivity of 3DGS performance to noise in the initial point clouds and total movements of Gaussians throughout the optimization process (Table 1,2) and show the limited transportability of Gaussians through visualizations (Figure 2).
> - **A novel practical and effective method:** By identifying the limitations of the original 3DGS optimization, we reveal simple modifications to encourage the transportability of the Gaussians. By introducing an efficient method that does not modify the entire optimization algorithm, our method can be applied to various 3DGS extensions that leverage the original 3DGS optimization in a plug-and-play manner.
>
> By the analysis of the 3DGS optimization and introduction of a practical and effective method to overcome the limitation of lack of Gaussian transportability, our method enables robust training of 3DGS with random point clouds.
>
> > While I appreciate their improvements to the original 3DGS methods, there are more variants of 3DGS, and I wonder would the same strategies proposed in this paper can improve the other 3DGS methods(such as scaffold-gs, etc).
> >
>
> We thank the reviewer for a highly valuable suggestion! Yes, our proposed strategy can be applied to various 3DGS extensions that leverage the optimization strategy of the vanilla 3DGS in a plug-and-play manner. However, the suggested Scaffold-GS[1] modifies the entire optimization strategy by defining a voxel grid from the provided SfM point clouds where the Gaussian parameters are learned using this voxel grid. This prevents our strategy from being directly integrated to its algorithm. Nevertheless, we observed that the accuracy of the initial point cloud is also highly important for Scaffold-GS, due to the quality of the initial voxel grid.
>
> As our strategy can also be seen as learning ideal point clouds together with the training of 3DGS, we observed that the positions of the learned Gaussians can be used as initial point clouds for other methods. Therefore, instead of directly integrating our strategy to Scaffold-GS, we have conducted an experiment of training Scaffold-GS with the point clouds achieved from Our strategy. Specifically,
>
> 1. We train 3DGS with random point clouds using our strategy. For time and memory efficiency, we train 3DGS for only 7000 steps. We also found that the number of Gaussians at 7000 steps are similar to the number of initial point clouds achieved with SfM which is around 7000 steps in the Mip-NeRF360 dataset[2].
> 2. We save the positions of the Gaussians as point clouds.
> 3. From the saved point clouds, we train Scaffold-GS for 30K iterations.
>
> The performance is shown below:
>
> | Methods | bicycle | flowers | garden  | stump | treehill | room | counter | kitchen | bonsai | Average |
> | --- | --- | --- | --- | --- | --- | --- | --- | --- | --- | --- |
> | Scaffold-GS (SfM) | 24.50 | 21.44 | 27.17 | 26.27 | **23.15** | **31.93** | 29.34 | 31.30 | **32.70** | 27.53 |
> | Scaffold-GS (Random) | 23.05 | 19.79 | 26.38 | 22.48 | 21.33 | 30.65 | 28.15 | 29.43 | 30.66 | 25.77 |
> | **Scaffold-GS + RAIN-GS (Ours)** | **25.32** | **21.82** | **27.75** | **26.45** | 23.06 | **31.93** | **29.57** | **31.82** | 32.21 | **27.77** |
>
> We observe that leveraging the point clouds achieved from RAIN-GS even outperforms the results of Scaffold-GS trained with SfM point clouds, successfully enabling Scaffold-GS to be trained from random point clouds. This reveals an additional application of our strategy. Even with accurate camera poses and point clouds achieved from SfM algorithms, RAIN-GS can further boost the performance of various 3DGS extensions.
>
> ---
>
> [1] Lu, Tao, et al. "Scaffold-gs: Structured 3d gaussians for view-adaptive rendering."
>
> [2] Barron, Jonathan T., et al. "Mip-nerf 360: Unbounded anti-aliased neural radiance fields."

---

> ### Author Response · Authors · 2024-11-24
> **Response to Reviewer pLoa (2/2)**
>
> > The method can improve 3DGS on the tanks-and-temples and Mip-NeRF360 dataset, I would like to see the effectiveness of this method on texture-less areas.
> >
>
> We would like to kindly direct the reviewer’s attention to the last row of Figure 4. The scene includes highly texture-less regions such as grass and the gray street, where our method outperforms the setting in both quality and quantitative comparisons (shown in scene57 of Table 4). Furthermore, in the additionally conducted experiments of reconstructing 3DGS with generated images from DimensionX [1] where consistent images are generated with a given camera pose, Ours show accurate reconstruction of texture-less regions such as the white wall in the top row of the newly added **Figure 5**. Note that SfM algorithms such as GLOMAP [2] fail to converge in these scenes highly due to the large texture-less regions making it hard to find accurate correspondences. In these scenarios, Ours largely outperform the reconstructed results of 3DGS trained from random point clouds.
>
> > The authors proposed the progressive low-pass filter to ensure 3D Gaussians receive more gradients during training. I wonder if is this equivalent to training 3D Gaussians from low-resolution images to high-resolution images in a coarse-to-fine manner. I would appreciate the authors to provide a fair comparison between the progressive low-pass filter and coarse-to-fine training strategy.
> >
>
> We thank the reviewer for mentioning an important point that is valuable for discussion. As mentioned by the reviewer, the progressive Gaussian low-pass filtering method is similar to the training of coarse-to-fine manner as large values of the low-pass filter makes the Gaussians to cover larger areas, resulting in a more blurry image (coarse image). Therefore, our approach can also be interpreted as a coarse-to-fine training strategy. However, our approach differs from simply training 3DGS from low-resolution images to high-resolution images as follows:
>
> 1. **More Gradient propagation:** One of the key motivation of encouraging the Gaussians to cover larger areas is to propagate sufficient gradients during training. This enables the Gaussians to sufficiently undergo cloning and splitting preventing the overfitting of Gaussians to a very local region.
> 2. **Supervision only from high-resolution image:** Unlike using low-res to high-res as supervision, the GT image in our strategy stays the same as a high-resolution image. This is beneficial as training 3DGS with different scales of images can cause aliasing problems as mentioned in Mip-Splatting[3].
>
>     We also conduct an experiment of replacing our progressive Gaussian low-pass filtering to training 3DGS from low-res to high-res images in the Mip-NeRF360 dataset [4]. For fair comparison, the experiments are done by following the schedules of our Gaussian low-pass filtering, where we simply replace the schedules to control the resolution of the target image.
>
>
> | Methods | bicycle | flowers | garden  | stump | treehill | room | counter | kitchen | bonsai | Average |
> | --- | --- | --- | --- | --- | --- | --- | --- | --- | --- | --- |
> | RAIN-GS (Resolution) | 25.039 | 21.778 | 26.985 | 26.882 | 22.511 | 30.675 | 28.551 | 31.220 | 31.552 | 27.244 |
> | **RAIN-GS (Ours)** | **25.373** | **22.118** | **27.277** | **27.029** | **22.887** | **30.866** | **28.681** | **31.416** | **31.610** | **27.473** |
>
> It can be observed that our strategy is more robust than training 3DGS from low-res to high-res images.
>
> ---
>
> [1] Sun, Wenqiang, et al. "DimensionX: Create Any 3D and 4D Scenes from a Single Image with Controllable Video Diffusion." arXiv preprint arXiv:2411.04928 (2024).
>
> [2] Pan, Linfei, et al. "Global structure-from-motion revisited." European Conference on Computer Vision. Springer, Cham, 2025.
>
> [3] Yu, Zehao, et al. "Mip-splatting: Alias-free 3d gaussian splatting." *Proceedings of the IEEE/CVF Conference on Computer Vision and Pattern Recognition*. 2024.
>
> [4] Barron, Jonathan T., et al. "Mip-nerf 360: Unbounded anti-aliased neural radiance fields." Proceedings of the IEEE/CVF conference on computer vision and pattern recognition. 2022.

---

> > ### Comment · Reviewer_pLoa · 2024-11-27
> > **Thanks for the rebuttal**
> >
> > I appreciate the authors for emphasizing their contributions. Most of my concerns are resolved, especially their experiments on training Scaffold-GS with the learned points with RAIN-GS. As for applying RAIN-GS on texture-less areas, I am impressed by the experiments on the RealEstate-10K dataset (I'm sorry to overlook this part) and the newly-added Fig.5. In a revised version, it could be better to highlight the challenges and differences (texture-less) of the RealEstate-10K dataset to other datasets (MipNeRF360, etc) in Sec.6.2. Overall, I think the method is valuable for 3DGS to scenes without rich textures. I appreciate the authors for their efforts on the detailed rebuttal and I would like to raise my score accordingly.

---

> > > ### Author Response · Authors · 2024-11-28
> > >
> > > We thank the reviewer for the positive evaluation of our work. We are delighted to find that our response have addressed the reviewer's concerns. We also appreciate all the valuable feedback from the reviewer, which helped us strengthen the contribution of the work. We further noticed that the reviewer has requested a quantitative evaluation of the trained camera pose and the evaluation can be found below:
> > >
> > > | Methods (w/ camera optimization) | Rotation (Degrees) $\downarrow$ | Translation (m) $\downarrow$ |
> > > | --- | --- | --- |
> > > | Random | 3.481 | 0.206 |
> > > | Ours | **2.001** | **0.146** |
> > >
> > > Aligned with the evaluation shown with PSNR, SSIM, and LPIPS metrics, Ours shows lower pose error when compared to the training with random point clouds. However, as the reviewer mentioned, we believe that other regularization and additional inputs (e.g., GT depth, GT Correspondence) should be further utilized to better mitigate the pose error problem.

---

> ### Author Response · Authors · 2024-11-25
> **More detailed explanations for our additional experiment**
>
> We would like to provide more detailed explanations for our additional experiment, where we replace the initial point cloud for Scaffold-GS with the point cloud from our method. This is only to clarify some of our initial explanations as suggested by reviewer gqEX.
>
> - **Training Iterations**
>
> The reported performance of Scaffold-GS with SfM points, random points, and points from RAIN-GS is the performance after training for 30K steps. Only RAIN-GS is trained for 7K steps.
>
> - **Initial point clouds**
>
> More initial points for point clouds or learned parameters for the Gaussians can provide extra performance boosts. Therefore, we have chosen 7K steps as the number of Gaussians after 7K steps is similar to the number of initial points from SfM. In addition, we only save the positions of the Gaussians as point clouds leaving other parameters unsaved. Note that this is even less beneficial compared to utilizing SfM point clouds as they also save the colors of the points, which are then used to initialize the SH coefficients of the Gaussians at the beginning of training.

---

> ### Author Response · Authors · 2024-11-26
>
> Dear Reviewer pLoa,
>
> As the author-reviewer discussion period is coming to an end, we wanted to kindly remind you of our responses to your comments. We greatly value your feedback and are eager to address any additional questions or concerns you might have.
>
> Please let us know if there's any further information we can provide to facilitate the discussion process.
>
> We are highly appreciated for your time and consideration.

---

### Official Review · Reviewer_y9Vo · 2024-11-04

**Soundness:** 2
**Presentation:** 2
**Contribution:** 2
**Rating:** 5
**Confidence:** 3

**Summary:**

This submission investigates the initialization strategy commonly used in existing 3D Gaussian Splatting (3DGS) optimization schemes. The authors present some technical insights towards the question of why the 3DGS optimizers tend to perform poorly when the centers of the 3D Gaussians are initialized from random values. They coin a new term in the paper and call the phenomena “low Gaussian transportability”. The authors also argue that the sole use of photometric error leads to the aforementioned limitation.

To address the observed limitations in optimizing 3DGS representations from noisier initialization, the authors have proposed some remedies, such as a) starting with fewer Gaussians with significantly larger covariances, applying low pass filtering progressively, introducing heuristics for reducing the gaussian size as the optimization progresses and the number of Gaussians in the scene increase, new splitting criteria.

Empirically these ideas lead to incremental improvements on the popular datasets (Mip-NeRF 360, RealEstate 10K, Tanks & Temples, Deep Blending) in terms of standard metrics (pSNR, SSIM, LPIPS, etc.). An ablation study was also reported to measure the benefit of each of the proposed modifications in isolation.

**Strengths:**

The authors propose certain modifications to the initialization steps prior to optimizing 3D Gaussian splatting representations. It leads to small but consistent improvements in the reconstruction quality on several datasets especially when the optimizer is started with noisy or random initialization of the 3D point coordinates. In general, the arguments and insights they present sounds reasonable in theory and seems to be helping in practice as shown in the empirical results.

**Weaknesses:**

The authors try to improve 3DGS performance when the Gaussians are initialized from random 3D points. While their results show some promise towards that end, they however assume that the camera poses are accurate. Typically, a full SfM pipeline must be executed to obtain accurate camera poses – must run bundle adjustment till completion. When running a full BA, accurate points are typically obtained as a by-product and no extra computation is needed. So, in other words, I don’t see a compelling situation where the 3DGS optimizer must be initialized from a random 3D point cloud when accurate camera poses are already available. In case of Re10K, the 3D points could be initialized using triangulated feature matches where both matching and triangulation could leverage the accurate poses.

However, it becomes much more interesting and relevant when camera poses are obtained from a different source and not necessarily extremely accurate.

While the paper is readable, the main paper lacks sufficient technical detail. The authors introduce the idea of limited Gaussian portability but in Section 4.2 there is an empirical study on the Mip-NeRF360 dataset without any deeper insights. Notably other authors have already investigated the issue of local minima in the optimization landscape. So I am not convinced that this paper extends the insight beyond what is already known in the literature. So I would argue against the first claimed contribution on page 2 “For the first time, we conduct an in-depth analysis and identify … “.

The ideas of improving the initialization seem reasonable but none of these modifications guarantees escaping local minima. For example, the authors state in Section 5.1 “To prevent the Gaussians from falling into local minima, we propose a simple yet effective modification to the original random initialization”. Even though some benefits of the proposed scheme are shown in the empirical evaluation, in the absence of any formal guarantees, the claims should be toned down a bit. Similarly, the specific split heuristic describes here (called ABE-Split) is a variation of existing splitting strategies and there is not much detail why this specific criteria should be used.

**Questions:**

I have a few comments ..

Typo:
page 2, were propose -> we propose

* Pan et al. 2024 accelerates the SfM pipeline but does not use learnable modules.
* In the abstract and in Section 1, the authors argue that solely using photometric loss is the main cause for poor optimization performance with noisy initialization. However, throughout the paper, the standard photometric loss continues to be used in the optimization objective without any modifications. Thus, it would help to put less emphasis on the particular point when motivating the problem.

**Details Of Ethics Concerns:**

No concerns.

---

> ### Author Response · Authors · 2024-11-24
> **Response to Reviewer y9Vo (1/4)**
>
> We thank the reviewer for the thorough review and valuable feedback that can improve our work when addressed. Our detailed response can be seen below:
>
> We summarized some of the reviewer’s words.
>
> > As accurate camera poses are obtained from a full pipeline of Bundle Adjustment such as Structure-from-Motion, I don’t see a compelling situation where the 3DGS optimizer must be initialized from a random 3D point cloud when accurate camera poses are already available. In case of Re10K, the 3D points could be initialized using triangulated feature matches where both matching and triangulation could leverage the accurate poses.
> >
>
> We agree that with accurate camera poses, a further post-processing of feature triangulation to achieve the point clouds is a possible direction to mitigate the issue of performance degradation of 3DGS from random point clouds. However, we would like to emphasize that the image pair-wise feature matching and triangulation is the main bottleneck of the SfM and BA algorithms which has a time complexity of $O(n^4)$ with respect to n input images[1]. Also shown in our Table 4, although Re10K[2] has an average 50 images per-scene which is about $1/4$ size of the Mip-NeRF[3] dataset, the processing time for COLMAP (Re10K : 6min, Mip-NeRF360: 19min) takes as much as the training time of 3DGS, which is a big bottleneck for settings where the camera poses are readily available.
>
> Our main contributions lie in removing the large dependency of 3DGS on accurate point clouds enabling the training with random points, which can largely reduce the processing time of BA even with accurate camera poses. In real-world scenarios, we find that there can be four outcomes of running SfM algorithms, in this response, we show that our method is effective and essential in all four settings:
>
> - SfM algorithms fail → providing no camera pose and point clouds
> - Camera poses are noisy
> - Camera poses are accurate but the point clouds include noise
> - SfM algorithms successfully converges
>
> Specifically, although SfM algorithms can fail to converge, the following settings provide camera poses from a different source:
>
> 1. **Reconstruction with generated images with text-to-3D generation models:** Recent generative approaches[4] have shown promising performance in generating consistent 3D images following a given camera trajectory as a condition prompt. The generated images are readily available with camera pose but without any point clouds. Here we show the results of training 3DGS with generated images using DimensionX[4], where Ours show significant improvements on PSNR over random point clouds. We also add qualitative results in Figure 5 of the revised pdf. It is worth noting that SfM algorithms [2] fail to converge in these scenes, making random point clouds the only choice for reconstruction.
>
>
>     | Methods / Scene | Scene1 | Scene2 |
>     | --- | --- | --- |
>     | Random | 11.17 | 12.27 |
>     | RAIN-GS (Ours) | **25.26** | **32.75** |
> 2. **Reconstruction with camera poses achieved from graphic rendering programs such as Blender:** Images generated from Blender is readily available with accurate camera poses and due to the fact the original 3DGS shows the result of NeRF Synthetic dataset[5] which is generated with Blender, from random point clouds. As random point clouds in [5] does not cause performance degradation due to the small scene that is bounded by the cameras, when dealing with larger scenes (e.g., Mip-NeRF360), applying Ours improve large performance over the random initialization setting. It has been shown that SfM algorithms fail in finding accurate correspondences in synthetic images, failing to converge [6].
> 3. **Reconstruction with camera poses achieved from sensors (Re10K):** As shown in Table 4, Ours achieve better performance than both SfM and random point clouds. We empirically find that with large texture-less regions, SfM algorithms can also fail to converge in these scenes.
>
> ---
>
> [1] Wu, Changchang. "Towards linear-time incremental structure from motion." 2013 International Conference on 3D Vision-3DV 2013. IEEE, 2013.
>
> [2] Zhou, Tinghui, et al. "Stereo magnification: Learning view synthesis using multiplane images." *arXiv preprint arXiv:1805.09817* (2018).
>
> [3] Barron, Jonathan T., et al. "Mip-nerf 360: Unbounded anti-aliased neural radiance fields." *Proceedings of the IEEE/CVF conference on computer vision and pattern recognition*. 2022.
>
> [4] Sun, Wenqiang, et al. "DimensionX: Create Any 3D and 4D Scenes from a Single Image with Controllable Video Diffusion." *arXiv preprint arXiv:2411.04928* (2024).
>
> [5] Mildenhall, Ben, et al. "Nerf: Representing scenes as neural radiance fields for view synthesis." *Communications of the ACM* 65.1 (2021): 99-106.
>
> [6] Chen, Yu, and Gim Hee Lee. "DReg-NeRF: Deep registration for neural radiance fields." Proceedings of the IEEE/CVF International Conference on Computer Vision. 2023.

---

> ### Author Response · Authors · 2024-11-24
> **Response to Reviewer y9Vo (2/4)**
>
> > It becomes much more interesting and relevant when camera poses are obtained from a different source and not necessarily extremely accurate.
> >
>
> We thank the reviewer for the thoughtful question. This question shows that our method can also be utilized when the camera poses are inaccurate. Accurate camera pose is one of the most important elements for high-quality novel view synthesis both for NeRF and 3DGS frameworks. To show the robustness of our framework with inaccurate camera poses, we have conducted experiments of training 3DGS with noisy camera poses from Mip-NeRF360 dataset[1], where the noisy camera poses are obtained by following the protocol of CamP[2], which adds approximately 5 degrees of noise to all the camera poses. Furthermore, as the original rasterizer of 3DGS does not propagate gradients to the camera poses, we have implemented a custom rasterizer for both 3DGS and ours to enable correction of the camera poses. We show both average results on Mip-NeRF360 dataset with and without camera pose correction below:
>
> | Methods (w/o camera optimization) | PSNR $\uparrow$ | SSIM $\uparrow$  | LPIPS $\downarrow$ |
> | --- | --- | --- | --- |
> | Random | 16.026 | 0.386 | 0.623 |
> | Ours | **16.492** | **0.404** | **0.607** |
>
> | Methods (w/ camera optimization) | PSNR $\uparrow$ | SSIM $\uparrow$  | LPIPS $\downarrow$ |
> | --- | --- | --- | --- |
> | Random | 18.916 | 0.453 | 0.572 |
> | Ours | **21.060** | **0.538** | **0.475** |
>
> We observe that although noise in the camera pose largely degrades the performance in the Mip-NeRF360 dataset, applying ours consistently achieve better results than the original random point cloud settings. The superior performance of Our method comes from the progressive Gaussian low-pass filtering and SLV initialization, where this approach is similar to the coarse-to-fine training adopted in settings of training noisy poses together with NeRF[3,4]. Combined with our experiments in Table1, Our approach can robustly train 3DGS in settings regardless of either the initial point cloud is noisy or the camera pose is noisy.
>
> > Notably other authors have already investigated the issue of local minima in the optimization landscape. So I am not convinced that this paper extends the insight beyond what is already known in the literature. So I would argue against the first claimed contribution on page 2 “For the first time, we conduct an in-depth analysis and identify … “.
> >
>
> We would like to clarify that although pixelSplat[5] has mentioned that the Gaussians can easily fall into local minima, our work extend their analysis as follows:
>
> - **In-depth analysis:** PixelSplat simply mentions the problem of Gaussians falling into local minima with the analogy of Gaussian optimization with the EM algorithm. They do not analyze how this limitation exhibits performance degradation. Unlike pixelSplat, we further identify that the optimization scheme of 3DGS lacks the transportability of Gaussians and show novel analysis qualitatively and quantitatively (Figure 1,2 / Table1,2) which has not been done before.
> - **With cloning and splitting:** One of the essential elements of 3DGS optimization is the adaptive density control which clones and splits the Gaussians during training. PixelSplat analyzes the problem Gaussian optimization without these cloning and splitting operation and explicitly mentions that “Gaussians falling into local minima is addressed when the cloning and splitting operation is possible”. However, we found that even with cloning and splitting, the Gaussians still lack transportability, falling into local minima.
>
> However, we agree with the reviewer that our analysis can be considered as an extension of pixelSplat and have toned down the expressions by removing “For the first time”.
>
> ---
>
> [1] Barron, Jonathan T., et al. "Mip-nerf 360: Unbounded anti-aliased neural radiance fields." *Proceedings of the IEEE/CVF conference on computer vision and pattern recognition*. 2022.
>
> [2] Park, Keunhong, et al. "Camp: Camera preconditioning for neural radiance fields." *ACM Transactions on Graphics (TOG)* 42.6 (2023): 1-11.
>
> [3] Lin, Chen-Hsuan, et al. "Barf: Bundle-adjusting neural radiance fields." *Proceedings of the IEEE/CVF international conference on computer vision*. 2021.
>
> [4] Yang, Jiawei, Marco Pavone, and Yue Wang. "Freenerf: Improving few-shot neural rendering with free frequency regularization." *Proceedings of the IEEE/CVF conference on computer vision and pattern recognition*. 2023.
>
> [5] Charatan, David, et al. "pixelsplat: 3d gaussian splats from image pairs for scalable generalizable 3d reconstruction." *Proceedings of the IEEE/CVF Conference on Computer Vision and Pattern Recognition*. 2024.

---

> > ### Comment · Reviewer_pLoa · 2024-11-27
> > **Experiments with noisy camera poses**
> >
> > I think the experiments with noisy camera poses are interesting. It is impressing to me the method can improve the performance significantly with camera pose optimization since there lacks existing methods to correct incorrect camera poses for 3D Gaussian Splatting (existing implementations work not good according to the [issue of GSplat](https://github.com/nerfstudio-project/gsplat/pull/123)). I'm also curious about the camera pose accuracy with camera pose optimization integrated in the experiments. Could the authors show more quantitative results on this experiment?

---

> > > ### Author Response · Authors · 2024-11-28
> > >
> > > We provide the quantitative results evaluating the pose error after optimization below:
> > >
> > > | Methods (w/ camera optimization) | Rotation (Degrees) $\downarrow$ | Translation (m) $\downarrow$ |
> > > | --- | --- | --- |
> > > | Random | 3.481 | 0.206 |
> > > | Ours | **2.001** | **0.146** |
> > >
> > > Aligned with the evaluation shown with PSNR, SSIM, and LPIPS metrics, Ours shows lower pose error when compared to the training with random point clouds. However, as the reviewer pLoa further mentioned, we believe that other regularization and additional inputs (e.g., GT depth, GT Correspondence) should be further utilized to better mitigate the pose error problem.

---

> > > > ### Author Response · Authors · 2024-11-29
> > > >
> > > > Dear Reviewer y9Vo,
> > > >
> > > > We have provided additional explanations and experimental results to make our contributions and work’s focus more clear. As the author-reviewer discussion period is coming to an end, we would appreciate it if the reviewer could take a look at our responses. We greatly value your feedback and are eager to address any further concerns or questions.

---

> ### Author Response · Authors · 2024-11-24
> **Response to Reviewer y9Vo (3/4)**
>
> Extended from our previous responses, we also find that our method can be effective in settings where the initial point clouds include noise or even when SfM algorithms have succeed in reconstructing the scene, providing accurate initial point clouds and camera poses.
>
> 1. **Initial point clouds include noise:**
>
> We found that our method can be effectively utilized when the initial point clouds include noise. Specifically, this can happen with large texture-less regions in the images.
>
> As shown in Table 1, adding a small noise $\epsilon$ sampled from a normal distribution of $N(0,0.5)$ directly degrades the performance of 3DGS (-0.4dB), due to its large dependency of accurate initial point clouds. In real-world applications, as the user cannot evaluate how accurate the point clouds achieved from SfM algorithms are, Our method can be applied to avoid unwanted performance degradation.
>
> 2. **SfM algorithms have successfully reconstructed the scene:**
>
> We found that our method can be effective even when SfM algorithms have succeed in reconstructing the scene, providing accurate initial point clouds and camera poses. Specifically, we have observed that our method can be interpreted as learning how the scene is reconstructed, together with the parameters of the 3D Gaussians through optimization. Therefore, the positions of the learned Gaussians can also be utilized as a point cloud, that further boosts the performance of 3DGS methods.
>
> To show the effectiveness of our approach, we have conducted additional experiments of replacing SfM point clouds with ours for training Scaffold-GS [1], which was suggested by reviewer pLoa. Specifically, we replace the initial point clouds with ours by the following steps:
>
> 1. We train 3DGS with random point clouds using our strategy. For time and memory efficiency, we train for only 7000 steps. We also found that the number of Gaussians at 7000 steps are similar to the number of initial point clouds achieved with SfM in the Mip-NeRF360 dataset[2] allowing fair comparison.
> 2. We save the positions of the Gaussians as point clouds.
> 3. From the saved point clouds, we train Scaffold-GS for 30K iterations.
>
> The performance is shown below:
>
> Note that Scaffold-GS (SfM), Scaffold-GS (Random), and Scaffold-GS + RAIN-GS (Ours) is all trained for 30K iterations and only the step of achieving point clouds from RAIN-GS is done for 7K iterations.
> | Methods | bicycle | flowers | garden  | stump | treehill | room | counter | kitchen | bonsai | Average |
> | --- | --- | --- | --- | --- | --- | --- | --- | --- | --- | --- |
> | Scaffold-GS (SfM) | 24.50 | 21.44 | 27.17 | 26.27 | 23.15 | **31.93** | 29.34 | 31.30 | **32.70** | 27.53 |
> | Scaffold-GS (Random) | 23.05 | 19.79 | 26.38 | 22.48 | 21.33 | 30.65 | 28.15 | 29.43 | 30.66 | 25.77 |
> | **Scaffold-GS + RAIN-GS (Ours)** | **25.32** | **21.82** | **27.75** | **26.45** | **23.06** | **31.93** | **29.57** | **31.82** | 32.21 | **27.77** |
>
> We observe that training Scaffold-GS from the point clouds achieved from our method yields improved performance, which reveals an additional application of our work. Even when SfM algorithms succeed to provide accurate camera poses and point clouds, various 3DGS methods can utilize the point clouds achieved from our method. Note that as our method starts training from random point clouds, this approach also enables other 3DGS methods to be trained from random point clouds.
>
> ---
>
> [1] Lu, Tao, et al. "Scaffold-gs: Structured 3d gaussians for view-adaptive rendering." Proceedings of the IEEE/CVF Conference on Computer Vision and Pattern Recognition. 2024.
>
> [2] Barron, Jonathan T., et al. "Mip-nerf 360: Unbounded anti-aliased neural radiance fields." Proceedings of the IEEE/CVF conference on computer vision and pattern recognition. 2022.

---

> ### Author Response · Authors · 2024-11-24
> **Response to Reviewer y9Vo (4/4)**
>
> > The ideas of improving the initialization seem reasonable but none of these modifications guarantees escaping local minima. For example, the authors state in Section 5.1 “To prevent the Gaussians from falling into local minima, we propose a simple yet effective modification to the original random initialization”. Even though some benefits of the proposed scheme are shown in the empirical evaluation, in the absence of any formal guarantees, the claims should be toned down a bit. Similarly, the specific split heuristic describes here (called ABE-Split) is a variation of existing splitting strategies and there is not much detail why this specific criteria should be used.
> >
>
> We would like to clarify that from our analytical findings about 3DGS optimization, the limitation of Gaussians falling into local minima leads to the lack of transportability, making large performance degradations when trained with random point clouds. Based on our analysis, our strategies are concentrated on largely improving the transportability of the Gaussians to mitigate the original limitation which results in large performance boost when trained with random point clouds. Although our methods could be designed in a more sophisticated manner as suggested by the reviewer, our main contributions were to reveal that even with very simple modifications to the original 3DGS optimization, we can largely improve the transportability of the Gaussians as shown in Table 13 of the revised pdf. This is important as existing 3DGS and its variations all assume that accurate initial point clouds are provided, which also exhibits large performance drops when initialized with random point clouds. To this end, our simple approach can be readily applied to these methods in a plug-and-play manner, orthogonally improving their performance when trained with random point clouds. However, we agree with the reviewer that our proposed method can sometimes exhibit outliers and cannot guarantee a 100% success in addressing the problem of Gaussians falling into local minima, and will further tone down our claims in our revised version.
>
> > Typo: page 2, were propose -> we propose
> >
>
> We apologize for the typo and have fixed all typos in our revised version.
>
> > Pan et al. 2024 accelerates the SfM pipeline but does not use learnable modules.
> >
>
> We thank the reviewer for pointing this out. We also acknowledge that GLOMAP[1], which replaces the incremental BA with a global BA accelerates the overall pipeline of SfM. However, we also find that it has been empirically found that GLOMAP provides less accurate reconstructions than COLMAP both in the issues of the public code and in our settings where GLOMAP fails to reconstruct images from the scene of Re10K[2] and DimensionX[3]. We would like to also emphasize that improved and accelerated methods for SfM can be used together with our approach. In real-world scenarios, as in our empirical findings of GLOMAP’s failure, the SfM algorithm can provide noisy camera poses or point clouds. In both settings, Ours can be applied to these noisy reconstructed outputs to robustly train 3DGS.
>
> > In the abstract and in Section 1, the authors argue that solely using photometric loss is the main cause for poor optimization performance with noisy initialization. However, throughout the paper, the standard photometric loss continues to be used in the optimization objective without any modifications. Thus, it would help to put less emphasis on the particular point when motivating the problem.
> >
>
> We thank the reviewer for this suggestion. We would like to again emphasize that one of our contributions to our method is by revealing simple modifications to the existing 3DGS optimization algorithm to enable the Gaussians to be robustly trained from random point clouds. Our modifications stem from our analysis of the limitations of the existing 3DGS optimization algorithm and as we propose a simple method to mitigate these issues, our method can be applied to various 3DGS extensions in a plug-and-play manner. However, we appreciate the reviewer’s suggestion and agree that tackling the problem of naive photometric loss optimization is an important problem that remains, and we will discover this direction as a future work.
>
> ---
>
> [1] Pan, Linfei, et al. "Global structure-from-motion revisited." *European Conference on Computer Vision*. Springer, Cham, 2025.
>
> [3] Zhou, Tinghui, et al. "Stereo magnification: Learning view synthesis using multiplane images." arXiv preprint arXiv:1805.09817 (2018).
>
> [4] Sun, Wenqiang, et al. "DimensionX: Create Any 3D and 4D Scenes from a Single Image with Controllable Video Diffusion." arXiv preprint arXiv:2411.04928 (2024).

---

> ### Author Response · Authors · 2024-11-26
>
> Dear Reviewer y9Vo,
>
> As the author-reviewer discussion period is coming to an end, we wanted to kindly remind you of our responses to your comments. We greatly value your feedback and are eager to address any additional questions or concerns you might have.
>
> Please let us know if there's any further information we can provide to facilitate the discussion process.
>
> We are highly appreciated for your time and consideration.

---

### Author Response · Authors · 2024-11-24
**General Response**

We are significantly grateful for all the reviewers thoroughly reviewing our manuscript and providing valuable feedbacks which can improve our work when addressed. We are also grateful that the reviewers recognize the strengths of our work, including the effectiveness of our strategy of largely improving the results of 3DGS with random point clouds (y9Vo, pLoa, Caqa, gqEX), the importance of the task (gqEX), analysis being intuitive (y9Vo, Caqa), impressive results with extensive experiments (pLoa,Caqa,gqEx) and the overall writing being well-written (Caqa).

Encouraged by the valuable feedbacks from the reviewers, we have conducted extra experiments which can verify the effectiveness and application of our approach. The revised pdf includes:

- Additional qualitative and quantitative results from generated images with text-to-3D models (Figure 6, Table 5)
- More detailed ablations of our components (Appendix F.1)
- RAIN-GS with other 3DGS extensions (Appendix F.2)
- RAIN-GS with noisy camera poses (Appendix F.3)
- Replacing progressive low-pass filtering with low-to-high resolution image training (Appendix F.4)
- Analysis of Movement of Each Gaussian in RAIN-GS (Appendix F.5)
- Additional analysis on Grendel-GS (Appendix F.6)
- Fixed typos

We have expressed the modified or newly added parts in red.

Furthermore, in this rebuttal, we have addressed key points raised by the reviewers, including:

- Where can RAIN-GS be effectively used? (y9Vo, Caqa)
- Can RAIN-GS improve other extensions of 3DGS? (pLoa)
- Additional comparisons and analysis (gqEx)

---

### Author Response · Authors · 2024-12-03
**Summary of our work**

Dear Reviewers,

We sincerely appreciate your valuable time and effort in reviewing our manuscript. As the author-reviewer discussion period has concluded, we would like to provide a final summary of our work.

Our work, **RAIN-GS**, addresses the critical question: "Why is 3D Gaussian Splatting (3DGS) largely dependent on the accuracy of the initial point cloud, even when provided with accurate camera poses?" Through our analysis, we reveal that this dependency stems from the limited transportability of Gaussians within the 3DGS framework. We further verify our analysis by proposing a simple yet effective modification that enables the Gaussians to transport further, largely improving the performance of 3DGS even with random point clouds.

We have demonstrated that RAIN-GS can be effectively applied in the following scenarios:
1. **Text-to-3D Generated Images** (Figure 5, Table 6)
2. **Images with Camera Poses Obtained from Sensors** (Figure 4, Table 4)
3. **Images with Noisy Camera Poses** (Appendix F.3)
4. **Joint Application with Other 3DGS Extensions** (Appendix F.2)

In scenarios 1 to 3, traditional Structure-from-Motion (SfM) algorithms often fail to converge, failing to provide accurate initial point clouds to the user. These scenarios verify the effectiveness and necessity of our work which can widen the application of 3DGS to real-world scenarios. Furthermore, the joint application of RAIN-GS with Scaffold-GS shown that applying our work can be beneficial even when point clouds from SfM is accessible, improving the performance of Scaffold-GS.

Thanks to all the valuable feedbacks from the reviewers, we have included several experiments in the Appendix that can strengthen and verify our contributions. These experiments and discussions will be incorporated into the main paper in the final version. We also thank the reviewers from acknowledging our responses and a positive evaluation of our work after discussion. We believe that our extensive experiments validate the effectiveness of RAIN-GS and its potential to significantly reduce the burden on users to obtain accurate initial point clouds, thereby broadening the practical applications of 3DGS.

Thank you once again for your thoughtful reviews and constructive feedback,

Authors of submission 6403.

---

### Meta-Review · Area_Chair_pBJv · 2024-12-20

**Metareview:**

This paper describes an initialization strategy for improving the quality of 3D Gaussian splatting-based novel-view synthesis. The strength of this paper is the improvement over the existing 3DGS initialization strategy. On the other hand, the downside is the technical contribution of the work. The proposed method is rather heuristic based on the empirical observation. More importantly, as one of the reviewer commented, if the accurate camera poses are obtained by structure-from-motion, we already have sparse tie points which can be used for initialization of Gaussians. While the authors responded on this respect saying that the feature matching and triangulation is the bottleneck, but for knowing camera poses, these steps are actually already included. The overall reviewer ratings were mixed around the borderline, and importantly none of the reviewers were excited about the paper. The AC weighed on the lack of the motivation for the designed initialization scheme and reached this decision.

**Additional Comments On Reviewer Discussion:**

There were a number of interactions between authors and reviewers particularly about experimental validation. I thank both of them for constructive discussions. Although the concerns related to the experiment was addressed, the motivation of the proposed initialization strategy was not still convincing because one could use the SfM tie points in the setting the paper considered.

---

### Decision · Program_Chairs · 2025-01-22

Reject